# Antimicrobial Effects of Metal Coatings or Physical, Chemical Modifications of Titanium Dental Implant Surfaces for Prevention of Peri-Implantitis: A Systematic Review of In Vivo Studies

**DOI:** 10.3390/antibiotics13090908

**Published:** 2024-09-23

**Authors:** Maria Gkioka, Xiaohui Rausch-Fan

**Affiliations:** 1Department of Dentistry, Division of Oral and Maxillofacial Surgery, Vaud University Hospital Center, 1005 Lausanne, Switzerland; 2Division of Conservative Dentistry and Periodontology, School of Dentistry, Medical University of Vienna, 1090 Vienna, Austria; xiaohui.rausch-fan@meduniwien.ac.at

**Keywords:** dental implants, titanium, peri-implantitis, antimicrobial, systematic review, in vivo studies, surface modifications, metal element coatings, bacterial colonization

## Abstract

**Introduction:** Peri-implantitis poses a significant challenge for implant dentistry due to its association with bacterial colonization on implant surfaces and the complexity of its management. This systematic review aims to assess evidence from in vivo studies regarding the antimicrobial efficacy of titanium (Ti) dental implant surfaces following physical/chemical modifications or the application of various metal element coatings in preventing bacterial growth associated with peri-implantitis. **Materials and Methods:** A literature review was conducted across four scientific databases (PubMed, Embase, Scopus, Web of Science), encompassing in vivo studies published between 2013 and 2024, and 18 reports were included in the systematic review. **Results:** The findings suggest that titanium dental implant surfaces, following physical/chemical modifications and metal element coatings, exhibit antimicrobial effects against bacteria associated with peri-implantitis in humans and various animal models. **Conclusions:** The reviewed studies indicated a reduction in bacterial colonization, diminished biofilm formation, and decreased signs of inflammation in the peri-implant tissues, which provides evidence that physical/chemical alterations on titanium dental implant surfaces or metal element coatings, like silver (Ag), zinc (Zn), magnesium (Mg), and copper (Cu), demonstrate antimicrobial properties in in vivo studies. However, caution is warranted when translating findings to clinical practice due to methodological disparities and high bias risks. Further larger-scale clinical trials are imperative to assess their long-term efficacy and validate their clinical applicability.

## 1. Introduction

Despite advancements in dental implantology, complications like peri-implantitis, caused by microbial infections, pose ongoing challenges. Dysbiosis, which disrupts microbial balance, contributes to peri-implantitis initiation [1,2,3]. Peri-implantitis, an inflammatory condition with biofilm formation and bone loss, can lead to implant failure [4,5,6,7]. If untreated, it may progress rapidly, often faster than periodontitis, and result in implant and prosthesis loss [6,7,8]. Managing peri-implantitis is challenging, as the efficacity of non-surgical protocols is limited. Non-surgical treatment usually focuses on establishing healthier soft tissue conditions before considering surgery [9,10].

Surgical treatment involves flap/resective approaches, cleaning the implant surface, and addressing bone defects. Despite numerous descriptions in the literature, long-term evidence for the effectiveness of surgical techniques and biomaterials in addressing peri-implantitis lesions is limited [9,11,12]. Once bacterial biofilms have developed, eliminating them becomes challenging, as the bacteria within these biofilms exhibit more excellent resistance to antibodies, phagocytes, and antibacterial drugs compared to planktonic bacteria [13]. Consequently, this complexity in the management of peri-implantitis explains why the prevention of microbial colonization of implants, especially in an early stage, is of high importance.

Surface treatments involve modification (physical, chemical, electrochemical, or combined) and coatings (physical, chemical, or combined) [14,15]. Common methods include sandblasting, plasma spraying, UV irradiation, anodization, acid-etching, and alkali-heat treatment [14,16,17,18,19,20]. Despite innovative strategies, there is a need for a comprehensive assessment of the evidence regarding the efficacy of these modifications.

Coatings aim to prevent bacterial adhesion, inhibit biofilm formation, or directly eliminate bacteria. Researchers explore diverse combinations of coatings to improve implants’ biocompatibility, bioactivity, and antibacterial properties [16]. Coatings are categorized as organic (e.g., growth factors, extracellular matrix (ECM) proteins) and inorganic (e.g., nanostructured calcium, hydroxyapatite, silver) [14,21,22]. Antibacterial coatings include drug-releasing, drug-free, and antibiofouling coatings [14]. Drug-releasing coatings release active molecules (antibiotics, antimicrobial peptides, antiseptics) through diffusion, solvent control, chemical control, or pH-sensitive mechanisms in a controlled way [14,23,24]. The release duration of the active molecule is influenced by the surface topographies of the implants and the amount of drug loaded [14,23,24,25]. Drug-free coatings, like antibacterial biopolymers and photoactive metal oxide nanoparticle coatings, avoid substance release, preventing associated toxicity [14,23]. Antifouling coatings, using hydrophilic polymers like poly(ethylene glycol), deter microorganisms [14,23]. Antibacterial metals and alloys exhibit long-term antibacterial ability, unaffected by wear or abrasion, providing a durable solution compared to other coatings [26]. The whole alloy exhibits antibacterial activity; therefore, despite any machining, implantation, wear or abrasion, corrosion, etc., no changes will be noticed regarding the antibacterial ability of these alloys. On the contrary, wear, abrasion, or machining will produce a fresh surface, enhancing antibacterial activity. However, for all the other antibacterial coatings, even when they include metals like Ag and Cu, any possible wear or abrasion will destroy the coating and significantly reduce antibacterial efficiency [26]. Additionally, metals and alloys offer ease of processing and are less likely to face issues related to drug resistance, making them a more suitable option for long-term antibacterial applications in this study [26].

Therefore, the purpose of this study was to investigate the antibacterial effect of these physical/chemical modifications of titanium implant surfaces as well as the antibacterial effect of the various metal element coatings, given the fact that these types of implant surface treatments are considered to aim for long-term effects. Considering that the existing literature, with a vast number of in vitro studies, has already proven the positive effect in vitro, the focus of this review will be placed on the in vivo animal and human studies.

In this context, the review will explore the key research question: do titanium dental implant surfaces, after physical/chemical modifications and modifications using various metal element coatings, exert an antimicrobial effect against bacteria associated with peri-implantitis, as demonstrated in in vivo studies?

## 2. Results

### 2.1. Animal and Human Models and Follow-Up

The research included fourteen studies employing five different experimental animal models—rats, mice, pigs, rabbits, and dogs—with sample sizes ranging from 3 to 45 animals [27,28].

In three studies [2,29,30], the implants were inserted in the dorsal area of rats and mice. In three other studies [28,31,32], the implants were incorporated into the femoral medullary cavity of rats. In the only study involving White Yorkshire Desi female pigs (50% crossbred), the titanium plates were mounted on acrylic and then attached to the pigs’ teeth [33]. In four studies, the implants were inserted in tooth extraction sites in Sprague–Dawley male rats, adult male beagle dogs, and Labrador dogs [27,34,35,36]. In one study, the implants were inserted into the bilateral femurs of rabbits, but the number of rabbits used was not mentioned [37]. In the study by Pan et al., the implants were inserted in the alveolar bone in front of the maxillary first molars [38]. In one study, implants were inserted into an incision in the rabbits’ back muscles [39].

Four studies [40,41,42,43] involved human experiments. Ionescu et al. recruited ten subjects, seven women and three men, aged 20 to 32 years old [42], whereas Assery et al. recruited four systemically healthy, nonsmoking adult subjects [43]. Cochis et al. involved seven subjects in their study, four males and three females, aged 20–27 years, with a mean age of 24 years [40], and Areid et al. recruited ten healthy, nonsmoking adult volunteers (six males, four females, aged from 25 to 56 years, with a mean age of 39.7 years) [41].

The observation period for the selected studies varied from 12 h [43] to 8 weeks [27,35].

### 2.2. Implant Surface Modification

One study by Mathew et al. specifically examined the antibacterial efficacy of micro-nano or nanoscale topographies created with a hydrothermal technique [33]. Sun et al. investigated the in vivo antibacterial results of the titanium oxide (TiO_2_) nanotube structure, after the anodic oxidation and annealing process, on selective laser melting (SLM) titanium substrates [27]. Four studies explored the antibacterial impact of metallic agents (Zn and Mg) incorporated into nano-engineered modifications applied to the implant surface [28,29,34,37]. More specifically, Wang et al. analyzed the antibacterial efficacy of zinc oxide (ZnO) nanospheres, ZnO nanorods, and a ZnO nanorod–nanosphere hierarchical structure (NRS) on Ti and titanium-zirconium (Ti-Zr) implant surfaces [29], and Yang et al. studied magnesium (Mg)-incorporated nanotube-modified titanium implants (NT-Mg) [28]. In addition, Wen et al. compared the in vivo synergistic effect of the ZnO nanoparticle-loaded mesoporous TiO_2_ coatings (nZnO/MTC-Ti) to pure titanium (Ti), mesoporous TiO_2_ coating-titanium (MTC-Ti), and ZnO nanoparticle-loaded-titanium (nZnO-Ti) [34]. Zhao et al. evaluated the bactericidal capability of the zinc oxide (ZnO) nanorods on Ti implants [37]. One study used Cu-doped TiO_2_ (TiO_2_-Cu) films combined with photothermal therapy to combat implant-associated infections [30]. Liu et al. and Wang et al., investigated the anti-infective ability of titanium-copper TiCu alloy [36,39]. Moreover, four studies, Jin et al., Kuehl et al., Yin et al., and Tran et al. reported on the antibacterial efficacy of metallic coatings, silver-zinc, silver, zinc-strontium, and selenium, respectively, on the implant surface [2,31,32,35]. In the study by Pan et al., a novel nanocoating was constructed on titania nanotubes, after anodization and annealing treatment, which was followed by a hydrothermal treatment to create SrTiO_3_/TiO_2_ nanotubes (STNT) and Al-STNT [38]. Piezo-sonocatalytic properties were expressed after an ultrasound treatment every two days with a power of 1.5 W cm^−2^ for 5 min, comprising, in total, seven treatments over a period of two weeks [38].

Regarding human studies, Ionescu et al. recruited ten subjects who wore an individual mandibular thermoformed acrylic customized tray with three half-implants fixed on the buccal side of the device, where the tested implant underwent laser-micro texturing in 136 s [42]. Assery et al. recruited four systemically healthy, nonsmoking subjects who wore a custom-made acrylic stent on the maxillary arch, where the tested implant underwent an extraoral surface decontamination with erbium-doped yttrium-aluminum-garnet (Er: YAG) at 2940 nm [43]. Cochis et al. recruited seven systemically healthy subjects with good oral hygiene who wore polyvinylchloride oral appliances with six specimens, where the tested specimens were treated by adding silver (Ag) and gallium (Ga) through electrochemical surface modification using the anodic spark deposition (ASD) method [40]. Areid et al. recruited 10 healthy volunteers who had four titanium discs attached to the buccal surfaces of their maxillary molars using flowable composite resin. The tested discs were modified with nanoporous titanium dioxide surfaces (TiO_2_), obtained using the hydrothermal (HT) coating method, and with UV light for 60 min under ambient conditions, using a 36 W puritec HNS germicidal ultraviolet lamp, with a dominant wavelength of 254 nm [41].

### 2.3. Implant Characteristics

The implants employed in the included studies exhibited diverse characteristics in terms of number, dimensions, and shape. According to the study, the total number of implants used varied between 9 [30] and 45 [28,33]. In certain studies [27,29,33,36,40,41,42,43], a design of within-subject comparisons was used, evaluating the effects of different conditions on the same set of participants, while in others [2,28,30,32,34,35,38,39], a between-subject design was used where distinct subjects were assigned to the test and control groups. In the studies of Tran et al. and Zhao et al., the number of implants used was not clarified [31,37]. In terms of shape, a range of configurations was observed, but a prevalent choice was the use of cylindrical implants, rods and oblong shape [2,28,30,32,33,34,35,36,37,38,39,42]. Wang et al. used foils/slices [29], whereas Assery et al., Cochis et al. and Areid et al. carried out the experiment with discs [40,41,43]. Tran et al. used both plates and screws for their in vivo experiment [31].

### 2.4. Infection Set-Up Procedure and Antibacterial Efficacy Test

Six studies explored the antimicrobial impact on *Staphylococcus aureus* (*S. aureus*) using various methodologies [2,28,29,31,32,37,39]. Apart from *Staphylococcus aureus*, Kuehl et al. and Tran et al. studied the antimicrobial impact on *Staphylococcus epidermidis*. *Porphyromonas gingivalis* was investigated in the studies by Wen et al. and Pan et al. [34,38], and *Streptococcus mutans* in the studies by Lu et al. and Areid et al. [30,41]. Pan et al., apart from *Porphyromonas gingivalis*, studied the antimicrobial effect on *Fusobacterium nucleatum* [38]. In the studies by Mathew et al., Ionescu et al., Assery et al., and Cochis et al., the native oral microbioflora was tested and there was no mention of any procedure for infection set-up [33,40,42,43]. In the study by Yin et al., 2-0 sutures were tied around the cervical area of the implant to induce peri-implantitis four weeks after implant insertion [35]. The same procedure was followed by the group in the study by Pan et al. [38]. Liu et al. chose an infection model by native microflora using a ligature and a sucrose-rich diet model [36].

Most studies employed the bacterial culture and colony counting method to assess antimicrobial efficacy [2,29,30,31,32,33,34,37,38,39,40,41]. In their investigation, Sun et al. used the next-generation 16S rRNA gene sequencing (NGS) technique to study the taxonomy of sampled oral bacteria and to conduct profiling and comparison of the expressed genes [27]. The same analysis was also used by Liu et al. [36]. In the study by Yin et al., X-rays and micro-CT examinations were performed, histological sections of soft tissues were analyzed with Gram and hematoxylin eosin staining, and immunofluorescence staining of CD3 completed the bacterial analysis [35]. Hematoxylin and eosin staining were also used for the histological analysis in the studies by Liu et al., Wang et al., and Pan et al. [36,38,39]. In the study by Tran et al., immunostaining was used to recognize methicillin-resistant *Staphylococcus aureus* (MRSA) and methicillin-resistant *Staphylococcus epidermidis* (MRSE) on the plates, utilizing rabbit anti-MRSA and mouse anti-Staph *epidermidis* antibodies [31]. In the same study, confocal microscopy imaging of the bacteria within the biofilms was performed [31]. Histological evaluation with Giemsa staining to evaluate bacterial distribution was executed in the studies by Yang et al., Jin at al., Zhao et al., and Pan et al. [28,32,37,38]. Assery et al. used multiphoton confocal laser scanning microscopy and Flu-oView FV1000 software to evaluate and capture the biofilm 3D structure and the live/dead bacteria ratio. Computational analyses of confocal biofilm images were carried out [43]. Ionescu et al. employed MTT (3-(4,5)-dimethylthiazol-2-yl-2,5 diphenyl tetrazolium bromide) assay, confocal laser scanning microscopy (CLSM), scanning electron microscopy (SEM), and energy-dispersive X-ray spectroscopy (EDS) to analyze the microbial burden and show the live/dead ratio of microbial cells [42]. Finally, Cochis et al. employed the colorimetric (2,3-bis-2-methoxy-4-nitro-5-sulfophenyl-5-((phenylamino) carbonyl)-2H-tetrazolium hydroxide) (XTT) assay to assess the bacterial metabolic activity [40].

### 2.5. In Vivo Antibacterial Results

Wang et al. showed that the presence of bacteria on samples modified with ZnO nanorods and ZnO nanospheres was lower compared to bare Ti or Ti-Zr implants, with samples featuring ZnO NRS modification demonstrating the most diminutive bacterial presence [29].

Accordingly, in the study by Yang et al., the authors used a radiographic analysis to evaluate the antimicrobial effect. Noteworthy radiographic indicators of advancing bone infection related to implants include periosteal reaction, osteosclerosis, osteolysis, and joint deformity. The authors have shown that the radiographic scores in the NT-Mg group were significantly lower than those in the Ti and NT groups from days 14 to 35 post-implantation (*p* < 0.01) [28]. At 35 days following surgery, the NT-modified group demonstrated slightly lower radiographic scores compared to the Ti group (*p* < 0.05), suggesting that the nanotubular structure alone exhibited some limited in vivo anti-infection potential [28]. Histopathological scores, including intracortical abscesses or inflammatory cells, destruction of cortical bone tissues, osteoclasts, and presence of bacteria were significantly higher in both the Ti and NT groups compared to the NT-Mg group (*p* < 0.01) [28].

In the study by Wen et al., the combined impact of MTC-Ti loaded with nZnO on nZnO/MTC-Ti allowed for controlled, prolonged release of Zn^2+^, effectively mitigating cytotoxicity risks associated with excessive release and avoiding the overproduction of reactive oxygen species (ROS) or induction of cell apoptosis [34]. In comparison to Ti, MTC-Ti also exhibited a modest resistance to bacterial activity, indicating the heightened inhibitory influence of the mesoporous structure on bacterial adhesion [34].

Zhao et al. demonstrated the bactericidal capability of the zinc oxide (ZnO) nanorods on Ti implants [37]. In the same study, the authors converted ZnO nanorods into hybrid zinc phosphate nanostructures, Zn_3_(PO_4_)_2,_ through a simple hydrothermal treatment, to mitigate the rapid degradation of ZnO nanorods and address the problem of cytotoxicity [37].

In the infection model described in the study by Liu et al., the TiCu implant sustained metabolic equilibrium and prevented the development of an acidic ecosystem. At the microbiota level, it maintained the balance between anaerobes and aerobes, thereby preserving the health of peri-implant tissues and preventing peri-implant diseases [36].

This was further supported in the study by Wang et al., where despite the introduction of the same quantity of *S. aureus* during the surgical procedure, the colonies count in the cp-Ti group was significantly higher than in the Ti-10Cu group (*p* < 0.01) [39].

Lu et al. investigated the combination of Cu^2+^ ions and photothermal therapy. Upon exposure to 808 nm near-infrared (NIR) light, they demonstrated that the TiO_2_-1Cu film, characterized by a low Cu content, exhibited effective in vivo antibacterial activity [30]. This outcome was attributed to the synergistic impact of both the photothermal effect and the presence of copper [30].

In the study by Mathew et al., the investigators showed that the commercial sand-blasted and acid-etched Ti (nano-polished) implant (COM) exhibited a microbial attachment exceeding 5 × 10^5^ colony forming units (CFU) cm^−2^. In contrast, the micro-nano textured Ti (SAN) and nano-textured Ti (TNL) implants displayed significantly lower microbial counts, measuring 1 × 10^5^ CFU cm^−2^ and 0.5 × 10^5^ CFU cm^−2^, respectively [33]. The superhydrophilic nature and heightened surface energy of SAN and TNL surfaces resulted in an impressive nearly 90% reduction in bacterial attachment in vivo, in contrast to the microscale surface of the COM implant [33].

A pivotal mechanism underlying the antibacterial activities of nanostructured titanium surfaces, as shown in Sun et al.’s study, appeared to involve the disruption of bacterial cellular membranes on the nanophase calcium phosphate embedded in the TiO_2_ nanotubes (NTN) surface [27]. No noteworthy distinctions were observed in terms of diversity and species richness among all the groups (Ti, NT, NTN) [27]. The in vivo antimicrobial efficacy of nanoporous TiO_2_ coatings was also described in the study by Areid et al., where plaque samples from the noncoated groups (UV-treated or not) exhibited a higher frequency of *S. mutans* presence in biofilms compared to the coated hydrothermal groups (UV-treated or not), with the number of colonized surfaces being seven and three, respectively [41].

Cochis et al. demonstrated a significant decrease in bacterial colonies on samples treated with silver Ag and gallium Ga, relative to the controls. Specifically, silver-coated samples exhibited a 30–34% reduction in bacterial colonies compared to the controls. Additionally, a marked decrease in microbial metabolic activity was noted in both gallium- and silver-treated specimens, with gallium-treated specimens showing the highest inhibition ratio (27–35%), as validated by colorimetric XTT analysis [40].

In the study by Jin et al., Zn and Ag were co-implanted into titanium by plasma immersion ion implantation (PIII). The Zn-PIII group displayed reduced bacterial growth on agar plates than the pure Ti group, and the Tryptic soy broth (TSB) culture appeared slightly less cloudy, indicating partial antibacterial efficacy. In contrast, both the Ag-PIII and Zn/Ag-PIII groups exhibited significantly reduced bacterial growth on agar plates, and the corresponding TSB cultures were negative, indicating a clear appearance and proving the outstanding antibacterial capabilities of Ag-PIII and Zn/Ag-PIII in vivo [32]. This exceptional antibacterial ability is attributed to the synergistic effects of the long-range interactions of Zn and the short-range interactions of Ag, resulting from the microgalvanic couples in the Zn/Ag co-implanted titanium [32].

Kuehl et al. demonstrated that the application of Ag coating demonstrates a preventive effect against *S. epidermidis* infection, with efficacy influenced by both inoculum size and duration of exposure, showcasing a more substantial impact during perioperative as opposed to postoperative infections [2]. While Ag coating exhibits a tendency to control perioperative *S. aureus* infections, it falls short of complete prevention [2]. Notably, the combined use of preoperative daptomycin (DAP) and Ag coating proves highly effective in preventing MRSA infections in vivo. The supplemental application of preoperative vancomycin (VAN) results in a significant reduction of planktonic growth and prevents adherence in 33% of cases [2]. Individually, preoperative DAP, VAN, or Ag-coated cages are insufficient to prevent persistent MRSA infections. Remarkably, when combined with preoperative DAP, Ag coating not only hinders the growth of planktonic cells but also prevents adherence, achieving a 100% prevention rate [2].

In another study, Yin et al. showed that implants engineered with a precise coating of Zn and Sr at a specific concentration (times that of 40 μM Zn^2+^ and 6 mM Sr^2+^) effectively mitigate the onset of peri-implantitis [35]. The control group exhibited a substantial bacterial presence in soft tissues, whereas the experimental group demonstrated notable antibacterial efficacy, impeding the progression to subsequent inflammatory stages [35].

In the study by Pan et al., a novel aluminum Al^3+^ doped strontium Sr^2+^ titanate/titanium dioxide nanotubes SrTiO_3_/TiO_2_ coating was employed as a nanocoating with ultrasound-reactive properties, showing some enhanced antibacterial efficacy after ultrasound treatment in vivo [38].

According to the study by Tran et al., confocal microscopy examination of bacteria within biofilms revealed that uncoated plates exhibited thick and densely packed layers compared to Se-coated plates. Se nanoparticle coatings displayed more individualized bacteria and bacterial aggregates, which were comparatively less dense [31]. The aggregates observed on Se-coated plates had a maximum thickness of approximately 3 μm, whereas biofilms on uncoated plates demonstrated maximal thickness ranging from 5 μm to 16 μm [31].

Ionescu et al. revealed in their investigation, after the use of a neodymium-doped yttrium aluminum garnet (Nd: YAG) source diode pumped solid state (DPSS) laser (355 nm wavelength), that machined and laser-treated surfaces exhibited lower colonization compared to those treated with grit blasting [42]. Remarkably, no notable distinctions were found between machined and laser-treated surfaces. However, this mitigating effect is compromised when the laser beam is directed at the titanium surface from a varied angle, particularly on the inclined section of the threads [42]. Finally, Assery et al. pointed out that the application of Er: YAG laser treatment on titanium implant surfaces does not yield a substantial impact on the initial formation of biofilms in the oral cavity [43].

## 3. Discussion

In the oral milieu, a multitude of over 500 microbial strains is responsible for both oral biofilm formation and the development of peri-implantitis [44]. Numerous investigations have affirmed an elevated likelihood of pathogen colonization on implant surfaces prone to susceptibility within the initial 6 h following surgery [44]. Consequently, ensuring effective antimicrobial actions on the inaugural day becomes imperative for achieving clinical success.

To prevent the increasingly widespread problem of peri-implantitis, various methods of implant surface modification have recently been suggested. Conventional methods like sandblasting and acid etching, while effective in reducing the osseointegration cycle, have a drawback: the modification to create a rough surface increases the risk of bacterial adherence, ultimately causing implant failure [35,45,46]. Incorporating bioactive materials to cover the surface, like metals, represents a groundbreaking approach to address these issues. The antimicrobial effectiveness of metals and alloys can be primarily attributed to the presence of ions of alloying elements, such as Ag^+^, Zn^2+^, and Cu^2+^ [35]. The prevailing understanding is that these metal ions eliminate bacteria by triggering the generation of ROS. ROS comprises oxygen reduction products like peroxides, superoxides, hydroxyl radicals (OH^−^), and singlet oxygen, which induces oxidative stress and results in the breakdown of bacterial cell membranes [26]. Inorganic antibiotic materials offer distinct advantages over traditional organic agents, including chemical stability, resistance to heat, and prolonged efficacy [47]. More recent findings suggest that the antibacterial properties of these metals and alloys are primarily governed by the formation of phases containing Ag or Cu rather than the release of Ag^+^ or Cu^2+^ ions [26,48].

When compared to organic coatings, the antibacterial metals and alloys exhibit sustained antibacterial efficacy because their properties are inherent to the entire alloy, ensuring that antibacterial effectiveness is maintained despite machining, implantation, wear, abrasion, or corrosion. Processes like wear and abrasion expose fresh surfaces, potentially enhancing the antibacterial activity. In contrast, for other antibacterial coatings, any possible wear or abrasion can damage or remove the coating, thus diminishing its efficacy significantly [26]. In addition to this, antibacterial metals and alloys present an ease of production and versatility, allowing for the fabrication of antibacterial materials in various forms (bars, sheets, complex shapes) using conventional metal processing techniques, without compromising their antibacterial properties [26]. Furthermore, these metal ions are recognized for their potent broad-spectrum antibacterial effects, and there is no risk of developing drug-resistant strains [26]. The focus of this review was on implant surface modifications that provide long-term, consistent antibacterial activity, which is why antibacterial metals and alloys were chosen over organic coatings.

Silver, as an inorganic antimicrobial agent, has historically been recognized for its antiseptic properties. Recent studies highlight the utilization of silver-embedded materials to combat bacterial infections linked to dental implants [49,50,51,52]. Furthermore, silver exhibits a broad spectrum of antibacterial activity and a reduced likelihood of bacterial resistance, without causing harm to mammalian cells at the specified concentrations [12,17,46,47,53,54,55,56]. The antibacterial effects of silver arise from the generation of ROS and the disturbance of membrane functionality [57]. The antibacterial properties of Ag have been demonstrated by the studies included in this systematic review. Specifically, Kuehl et al. have shown that Ag-coated titanium-aluminum-niobium (TiAlNb) alloy prevented *S. epidermidis* infection in a manner that depends on both the inoculum size and duration of exposure, especially when combined with antibiotic prophylaxis [2]. The combined effect of preoperative DAP and Ag coating proved highly effective at inhibiting the growth of planktonic cells and preventing adherence, achieving a 100% prevention rate for MRSA infections in vivo. The additional impact of preoperative VAN resulted in a notable decrease in planktonic growth and prevented adherence in 33% of cases [2]. Moreover, the plasma immersion ion implantation of Ag (Ag-PIII) has revealed an excellent in vivo antibacterial effect in the study by Jin et al. [32].

There is a high level of confidence that the presence of Zn enhances the bactericidal effectiveness of titanium implants [26,58,59]. The primary mechanism of antibacterial action following the electrochemical reactions of ZnO nanoparticles with body fluids is commonly attributed to the release of Zn^2+^ ions. However, the optimal concentration, defined as the level at which the maximum bactericidal activity occurs and where higher concentrations may diminish the desired effect, remains undetermined, as shown in the systematic review and meta-analysis of Silva Lima Mendes et al. [58]. The antibacterial effect of Zn was also confirmed in the in vivo studies included in the current systematic review. In the study by Wen et al., the combined influence of MTC-Ti and ZnO nanoparticles on the nanocomposite material was investigated. In vivo, the MTC-Ti structure in nZnO/MTC-Ti led to prolonged antibacterial success by controlling the release of Zn^2+^. The continuous release of Zn^2+^ established a durable antibacterial environment, resulting in sustained inhibitory effects. This approach prevented the occurrence of excessive cytotoxicity due to Zn overdose without eliciting heightened levels of ROS or inducing cell apoptosis [34]. In addition, the experimental findings in the study by Wang et al. demonstrated that the dual-layered structures of Ti ZnO nanorods−nanospheres hierarchical structure or Ti-Zr ZnO nanorods–nanospheres hierarchical structure exhibited significant antibacterial efficacy in vivo, maintained good stability, and showed low toxicity [29,60]. However, the results should be cautiously evaluated as, in this study, a dental infection model was not created, and the direct bonding strength of the double-layered ZnO nanorod–nanosphere structures to the substrate surface was not confirmed using appropriate instruments [29]. The bactericidal efficacy of ZnO nanorods was additionally described in the study by Zhao et al., who employed a hydrothermal treatment in order to create a hybrid zinc phosphate nanostructure (Zn_3_(PO_4_)_2_) and overcome the problem of cytotoxicity by modulating Zn^2+^ release [37]. Furthermore, Yin et al. employed micro-arc oxidation (MAO) technology in their study to create a Zn/Sr experimental implant coating model. Specifically, the in vitro component of their study showed that the ideal concentration of the two ions in the electrolyte should be six times that of 40 μM Zn^2+^ and 6 mM Sr^2+^, thus creating the Zn/Sr implant MAO-6. Implants treated with this specific concentration of Zn and strontium Sr coating have proven effective in preventing peri-implantitis in vivo [35].

New methods, such as ultrasound therapies and antimicrobial nanomaterials, have shown promise in successfully eradicating biofilms. The antibacterial effect is maximized through the synergistic action of ultrasound treatment and different nanocoatings [59]. In this systematic review, the ultrasonic-stimulated piezoelectric effect was described in the study by Pan et al. where doping SrTiO_3_ with Al^3+^ ions caused crystal distortion and enhanced sonocatalytic efficiency through oxygen vacancies and piezoelectric properties. This led to increased ROS generation, induced by ultrasound, making Al-doped strontium titanate titania nanotubes (Al-STNT) a promising candidate for antibacterial applications [38].

Copper effectively eradicates a wide range of bacterial and fungal species [17,61,62,63]. The primary mechanism underlying its antibacterial efficacy involves damaging respiratory enzymes and disrupting bacterial membrane integrity, leading to biochemical disarray, cell lysis, or even cell death. However, the response of cells to Cu^2+^ ions is concentration-dependent due to potential cytotoxicity [62]. In the in vitro study by Li et al., a novel biomimetic coating featuring micro/nanoscale topography and antibacterial Cu was developed for Ti-based implants using a hybrid approach involving plasma electrolytic oxidation (PEO) and hydrothermal treatment [64]. This implant coating containing Cu-TiO_2_ exhibited reduced bacterial viability compared to the control group, confirming its antibacterial properties [64]. The Cu antibacterial action was further confirmed in the in vivo study by Lu et al., who explored the combination of Cu^2+^ ions and photothermal therapy, finding that the TiO_2_-1Cu film with a low Cu content exhibited effective in vivo antibacterial activity when exposed to 808 nm NIR light [30]. This outcome was attributed to the synergistic impact of both the photothermal effect and the presence of Cu. In the same study, Ti and TiO_2_-0Cu groups showed substantial bacterial populations on agar plates, while the TiO_2_-1Cu group displayed significantly fewer bacteria, achieving an impressive antibacterial efficiency of 96% [30]. This was further supported by the study by Liu et al., who indicated that the histological score for the TiCu group was significantly lower than that of the Ti group, underlying the TiCu implant’s superior anti-infection properties. In the same study, the anti-infection mechanism was investigated by employing 16S rRNA gene and metagenomic sequencing [36]. In addition, as demonstrated in another study by the same group, Wang et al., numerous colonies were observed in the Ti group at all time points. However, in the Ti-10Cu group, only a few bacterial colonies were detected 1–4 days post-implantation, and no bacterial colonies were found after 7 days post-implantation [39].

Magnesium showcases antibacterial effects attributed to the alkaline pH generated during its degradation, but this might not be adequate in dynamic environments like the human body [65,66,67]. The antimicrobial properties of corroding Mg, observed in vitro and in certain in vivo scenarios, are associated with increases in pH [65,66]. The absence of similar effects in vivo might be attributed to the possibility that the pH increases in the peri-implant tissue were not significant enough to promote the antimicrobial effect of corroding Mg and counteract bacterial colonization [66]. Another consideration is that precipitates on the Mg corrosion layer could potentially act as a barrier, limiting the diffusion of alkaline corrosion products [66]. A fundamental mouse model was established in the study by Imran Rahim et al. where, contrary to expectations, bacteria successfully colonized Mg implants, forming distinct, densely populated, persistent, and antibiotic treatment-resistant biofilms [66]. In the study by Yu et al., Zn/Mg ions were co-implanted onto titanium through plasma immersion ion implantation, resulting in the creation of Zn/Mg-PIII [68]. The in vitro antibacterial effect was more likely attributed to the presence of implanted Zn rather than to Mg presence. Bacterial counting results indicated that Zn/Mg-PIII and Zn-PIII surfaces exhibited significantly higher inhibitory rates against the growth of oral anaerobes (*Porphyromonas gingivalis*, *Fusobacterium nucleatum*, and *Streptococcus mutans*) compared to Mg-PIII [68]. The highly mobile and cytotoxic nature of metal ions released by the coatings, along with their potential to enter living cells in high concentrations and harm healthy cells, poses a challenge [69]. A proposed solution involves the formation of nanotube patterns on the substrate surface, incorporated into the structures to achieve a controlled and sustained antibacterial effect through gradual release [69,70]. In the context of this systematic review, Yang et al. assessed the antimicrobial effect of magnesium-incorporated nanotube-modified titanium implants using histopathological and radiographic analysis [28]. A substantial presence of bacteria, particularly in the Ti group, was observed within the cortical bone tissues and intramedullary cavities, as indicated by Giemsa staining. Furthermore, there was a notable increase in active osteoclasts, along with the progression of bone infection, in both the Ti and NT groups in comparison to the NT-Mg group, as evidenced by TRAP staining [28]. Radiographic indicators of advancing bone infection related to implants, such as periosteal reaction, osteosclerosis, osteolysis, and joint deformity, were noted. Significantly lower radiographic scores in the NT-Mg group compared to the Ti and NT groups were observed from days 14 to 35 post-implantation (*p* < 0.01) [28]. At 35 days post-surgery, the NT-modified group showed slightly lower radiographic scores compared to the Ti group (*p* < 0.05), indicating that the nanotubular structure alone demonstrated limited in vivo anti-infection potential [28]. The nanotubular structure alone was insufficient to combat severe implant-related bone infection effectively [28]. The antimicrobial efficacy of nanotube-modified titanium was also demonstrated in the study by Sun et al., where the disruption of bacterial cellular membranes on calcium phosphate nanoparticles embedded in TiO_2_ nanotubes could represent a crucial mechanism underlying the antibacterial properties of nanostructured titanium surfaces [27]. Furthermore, TiO_2_ nanotubes can modulate the immune response of macrophages. Host immune responses and inflammatory conditions may also impact the intricate interaction between the host and biofilm, consequently influencing the composition and functions of the colonizing microbiota [27]. Areid et al. also showed that hydrothermally induced nanoporous TiO_2_ surfaces successfully inhibited *S. mutans* adhesion and reduced biofilm development when compared to non-treated titanium surfaces [41]. In the same study, UV light treatment endowed titanium surfaces with antimicrobial properties and exhibited a tendency towards less microbial adhesion when contrasted with non-UV-treated titanium surfaces [41].

Gallium antibacterial activity has been demonstrated in in vitro studies, like the one by Yamaguchi et al., where the gallium-treated Ti specimen exhibited high antibacterial efficacy, as shown by the confocal microscopy images of the Ti surfaces containing multi-resistant *Acinetobacter baumannii* (MRAB12) [71]. In our systematic review, Cochis et al. observed a notable reduction in bacterial colonies on samples treated with Ag and Ga when compared to the control untreated groups in both colony-forming unit CFU and colorimetric XTT assay. In particular, silver-coated samples showed a decrease of 30–34% in bacterial colonies relative to the controls. Furthermore, a significant decline in microbial metabolic activity was detected in both gallium- and silver-treated specimens, with the gallium-treated specimens demonstrating the highest inhibition ratio (27–35%) [40].

Mathew et al. evaluated the performance of distinctive micro-nano or nanoscale topographies, achieved through a simple hydrothermal technique, in both short and long-term scenarios using a split-mouth design within a porcine model, both before and after loading [33]. They managed to demonstrate the effectiveness of nano/micro-nano textured designs for clinical use and provide evidence supporting their enhanced osseointegration and decreased bacterial adhesion. Specifically, they showed that, in vivo, surfaces with superhydrophilic characteristics and high surface energy, such as micro-nanotextured Ti and nanotextured Ti, exhibit a reduction of nearly 90% in bacterial attachment compared to the microscale surface of commercially available sandblasted and acid-etched Ti (nano polished) [33].

The application of nanoscale surface patterning methods for creating various nanopatterns like ordered stripes, pits, pillars, or squares, is currently a subject of significant interest. A study by Narendrakumar et al. focused on coating titanium surfaces with TiO_2_ nanotubes using an anodization process. These anodized nanostructures exhibited antibacterial properties influenced by their diameter and contact angle [15,63,72]. Specifically, studies have demonstrated that smaller diameter nanotubes (15 nm) maximize the induction of these cellular processes, while larger diameter tubes (100 nm) hinder them, potentially leading to cell death and apoptosis [73]. Studies have reported that nanomaterials display antibacterial properties due to various biophysical interactions between the nanoparticles and bacteria. Notably, metallic nanomaterials like silver, gold, copper, and titanium are favored for their specific physicochemical properties, which contribute to their potent antibacterial effects [57,74]. When considering nanoparticles, nano-ZnO was found to be the most toxic at concentrations below one milligram per liter, followed by nano-CuO and finally, nano-TiO_2_. Altering the shape and length of the material can help mitigate toxicity, making it an effective bactericidal agent that is safe for the human body [57,75]. As per Tran et al.’s research, examining bacteria within biofilms through confocal microscopy revealed that plates coated with selenium nanoparticles (SeNPs) exhibited more dispersed bacteria and bacterial aggregates with lower density than those without coating, displaying dense and thick layers [31]. On the coated Se plates, observed aggregates had a maximum thickness of around 3 μm, while biofilms on uncoated plates showed maximal thickness ranging from 5 μm to 16 μm [31]. The positive effect of Se nanoparticles was described as well by Liu et al. in their in vitro study, where their findings indicated that the inclusion of SeNPs in titania nanotubes (TNT) samples (TNT-Se) hindered the growth of both *Escherichia coli* and *Staphylococcus aureus* when compared to unaltered TNTs. At their peak efficacy, TNT-Se samples decreased the density of *E. coli* by 94.6% and *S. aureus* by 89.6% in comparison to titanium controls [76].

Studies have established the effectiveness of laser irradiation in promoting soft tissue wound healing, stimulating osteoblast proliferation, and facilitating bone healing. In the context of peri-implantitis treatment, the laser directly interacts with the implant surface [45,77,78,79]. However, there is scarce information regarding the impact of titanium surface modification resulting from laser use on bacterial adherence. In their study, Ionescu et al. used a Nd: YAG source diode pumped solid state (DPSS) laser (355 nm wavelength) and discovered that surfaces treated through machining and laser techniques demonstrated reduced colonization compared to those subjected to grit blasting [42]. However, the optimal manifestation of this effect occurs when the surface is exposed directly to the laser beam in an orthogonal manner [42]. Assery et al. highlighted that the application of Er: YAG laser treatment on titanium implant surfaces does not have a substantial impact on the initial formation of biofilms in the oral cavity, despite the fact that the surface that emerged after Er: YAG laser treatment displayed visible changes on titanium discs [43]. In contrast, Schwarz et al. found that Er: YAG irradiation (100 mJ/pulse, 10 Hz) did not result in observable alterations on different titanium surfaces [80]. In a separate study, Kreisler et al. showed that Er: YAG irradiation (60 mJ/pulse, 10 Hz) effectively eliminated bacterial cytotoxic components from implant surfaces in vitro, maintaining the surface morphology of microstructured surfaces unchanged [81]. The variations in irradiation parameters, such as contact mode, water irrigation, and irradiation angle and time, pose challenges in comparing outcomes across different studies [43,45]. Furthermore, Assery et al. applied the laser to discs rather than implant bodies. Taking into consideration the structural distinctions between these components, such as the presence of threads and grooves, as well as the fact that only the early biofilm formation was assessed in this study, varied outcomes may be easily reached [43].

### Limitations to Consider

Current research, exemplified by Wang et al., has highlighted the presence of anaerobic bacteria in peri-implantitis, but the efficacy of nanostructured titanium in combating these bacteria remains uncertain [29]. Real-life assessments are challenging because of the complexity of oral biofilms and their interactions with the host. Most researchers used single-pathogen infection models to evaluate anti-infective implants, overlooking the impact of the oral microbiome on peri-implant infections [36]. To address this gap, innovative research methodologies are necessary for comprehensive microbiome analysis in in vivo settings to understand the impact of nanostructures on bacterial communities. Sun et al. not only explored the taxonomy of sampled oral bacteria but also conducted profiling and comparison of the expressed genes, delving into bacterial functions on various nanostructured surfaces [27]. Next-generation sequencing for studying the microbiome provides a greater understanding of potential causal relationships between bacteria and disease, extending beyond culturable or cultivatable species [36,82,83].

Discrepancies in experimental designs, reporting, and limited sample sizes in in vivo animal studies impact the ability to compare outcomes and hinder the execution of a meta-analysis. Host variations and immune responses can further influence the intricate host-biofilm interaction, affecting the composition and functions of colonizing microbiota, demanding careful interpretation and cautious application of results [29].

While human studies are crucial for assessing clinical effectiveness, animal studies may offer insights into the biological and biomechanical aspects in a more controlled experimental setting, providing essential mechanistic and preclinical data. Ethical considerations and limitations in human studies make animal studies valuable for addressing specific research gaps. Combining evidence from both types of studies is essential for a comprehensive understanding of how surface modifications on dental implants affect bacterial colonization and related outcomes. However, caution is advised in interpreting findings due to variations in experimental methods and the high risk of bias in many studies. Future research should focus on larger clinical investigations to validate the antibacterial efficacy of titanium implants. Lastly, the potential for overlooked pertinent articles based on search criteria should not be neglected, as it may introduce an element of uncertainty regarding the results of this review.

## 4. Materials and Methods

### 4.1. Protocol Development

This review was reported in accordance with the PRISMA (Preferred Reporting Items for Systematic Review and Meta-Analyses) statement [84,85]. Prior to commencing the systematic literature review, the PICO question was stated:PICO question:P (Participants/Test group): Titanium dental implants in human/animal models.I (Intervention): Ti surfaces after physical/morphological modifications and/or modifications using various metal elements.C (Control group): Ti surfaces without modifications or coatings.O (Outcome): in vivo antimicrobial effect.

### 4.2. Search Strategy

A search was carried out in four electronic databases (PubMed, Embase, Scopus, and Web of Science) between 2013 and 14 July 2024, to identify all relevant articles. The search was limited to studies published in the last ten years. This was done to ensure that this review reflects the most current research and advancements in the field. More precisely, the search was conducted using the following terms:PubMed:

(dental implants) AND (titanium) AND (antimicrobial OR anti-microbial OR bactericidal OR antibacterial OR anti-bacterial OR bacteriostatic)) AND (humans OR animals) AND (2013:2024[dp])

Embase:

(‘dental implants’/exp OR ‘dental implants’ OR ((‘dental’/exp OR dental) AND (‘implants’/exp OR implants))) AND (‘titanium’/exp OR titanium) AND (‘antimicrobial’/exp OR antimicrobial OR ‘anti-microbial’ OR bactericidal OR antibacterial OR ‘anti-bacterial’ OR bacteriostatic) AND (‘in vivo study’/de OR ‘human’/de OR ‘animal’/de) AND [2013–2024]/py AND ‘article’/it

Scopus:

dental AND implant AND titanium AND (antimicrobial OR anti-microbial OR bactericidal OR antibacterial OR anti-bacterial OR bacteriostatic)) AND PUBYEAR > 2012 AND PUBYEAR < 2025 AND (LIMIT-TO (SUBJAREA, “DENT”)) AND (LIMIT-TO (DOCTYPE, “ar”)) AND (LIMIT-TO (LANGUAGE, “English”)) AND (LIMIT-TO (EXACTKEYWORD, “Dental Implants”) OR LIMIT-TO (EXACTKEYWORD, “Human”) OR LIMIT-TO (EXACTKEYWORD, “Animal”)

Web of Science:

TS = (dental AND implant * AND titanium AND (antimicrobial OR anti-microbial OR bactericidal OR antibacterial OR anti-bacterial OR bacteriostatic)) AND TS = (human * OR animal*) AND PY = (2013–2024)

### 4.3. Study Inclusion/Exclusion Criteria

This systematic review included all in vivo human and animal studies from the last ten years assessing the antibacterial effect of either metal coatings and/or any morphological/chemical modifications of dental implant surfaces.

Exclusion criteria for all studies were as follows: articles not written in English; reviews, expert opinion articles, and conference proceedings; articles assessing only dental implant osseointegration; articles studying implant surface modifications for the treatment of periodontitis and biofilm removal; articles referring to non-metal coatings like proteins/peptides, antibiotics, chemotherapeutics, bioactive glass, bioactive ceramic (hydroxyapatite), polymers and biodegradable elements, etc.; and any other irrelevant articles related to the broader field of dentistry or implantology. During the screening process, several studies were identified that pertained to other aspects of implantology, rather than the antibacterial properties of implant surfaces. Additionally, articles related to other domains of dentistry, like prosthetics, endodontics, and orthodontics, were also excluded because they did not align with the specific research question about peri-implantitis prevention. The titles and abstracts for all search results were reviewed. Full-text articles were obtained to determine studies that met inclusion and exclusion criteria.

### 4.4. Identification and Study Selection

The electronic database search on 14 July 2024 identified 2161 articles. Two additional articles were identified through manual search. After duplicate removal (n = 402), all remaining titles and abstracts were screened for relevance. The study selection process was conducted by two independent reviewers. Both reviewers individually assessed the titles and abstracts of all retrieved studies to determine their eligibility based on the inclusion and exclusion criteria. In cases where there was disagreement or uncertainty regarding the inclusion of a study, consensus was reached through discussion between the two reviewers. Through title screening, 1565 articles were initially deemed irrelevant per the inclusion and exclusion criteria. 196 articles that met the inclusion and exclusion criteria were sought for retrieval. Eight full texts could not be retrieved, and an eligibility assessment for these was carried out based on the abstracts. These 8 articles, for which the full text could not be retrieved, were excluded after an abstract evaluation, as they did not fulfill the inclusion criteria. After the eligibility assessment of all 196 articles, 18 reports were selected and included in the systematic review. The full text was retrieved for all 18 reports. The 178 excluded articles were either exclusively in vitro studies or combined in vivo and in vitro studies that did not assess the in vivo antibacterial effect, or concerned in vivo studies that investigated hybrid coatings. Meta-analysis was not conducted due to the scarcity and heterogeneity of the studies.

The study selection process, as previously described, is illustrated in Figure 1.

### 4.5. Data Extraction

For data extraction, one investigator was responsible for collecting the relevant data from each included study. To ensure the accuracy and reliability of the extracted data, the second investigator performed a thorough check and control of the data.

The information from the 18 studies included is presented in Table 1 and Table 2. Due to the predominantly descriptive nature of the existing literature, additional statistical assessments could not be conducted.

### 4.6. Risk of Bias and Assessment of Quality for the Selected Studies

The clinical human studies included in the analysis were evaluated using the revised Risk of Bias (RoB) assessment tool, which follows the Cochrane RoB assessment method outlined by Higgins and colleagues in 2019, as demonstrated in Figure 2 [86,87]. For preclinical animal studies, an assessment tool created by the Systematic Review Centre for Laboratory Animal Experimentation (SYRCLE) was used (Table 3). This RoB tool is an adaptation of the Cochrane tool, tailored to address biases specific to animal studies, as detailed in the work by Hooijmans and others [88]. In addition, the ARRIVE guidelines ensured quality assessment of the animal studies, which stands for Animal Research: Reporting of In Vivo Experiments, as shown in Table 4. These guidelines provide a checklist of recommendations to ensure comprehensive and transparent reporting of research that involves animals [89,90]. The assessment and scoring of study quality based on the ARRIVE criteria resulted in an average score of 16. Most of the included studies did not give any information on the procedure of allocating animals into different experimental groups, nor did they provide any information regarding the conditions in which animals were kept during the experiment and the overall care provided to ensure their well-being. No study mentioned any adverse events, and only one assessed the generalizability of their findings to evaluate their relevance to the human population.

## 5. Conclusions

Modern surface treatment technologies, particularly those incorporating metal ions like Ag^+^, Zn^2+^, and Cu^2+^, exert long-lasting antimicrobial ability and show promise in combatting the growing challenge of peri-implantitis. Until now, the evaluation of the antibacterial properties of biomaterials used in oral implants has heavily relied on diverse in vitro experiments. However, these outcomes may only reflect how materials affect specifically tested pathogens, lacking insight into the broader biological consequences on complex oral microbiomes and interactions between the host and biofilm. Moving from laboratory experiments to broad-scale production and widespread clinical application encounters difficulties for several reasons. Specifically, there is significant variability in the methodologies used across different studies, making it challenging to compare results and establish standardized protocols. Differences in bacterial strains, experimental conditions, and measurement techniques can lead to inconsistent findings. Furthermore, antibacterial coatings must demonstrate not only efficacy but also long-term safety in humans, which requires extensive preclinical and clinical trials. In addition, the cost of manufacturing implants with antibacterial coatings may be prohibitive, especially if the techniques require specialized equipment or materials that are not readily available. Therefore, the development and implementation of new surface treatments remain challenging. Moreover, while many studies demonstrate initial antibacterial efficacy, there is a need for long-term data to ensure that these effects are durable over the lifespan of the implant and under the dynamic conditions of the oral environment.

In exploring the diverse landscape of implant physical/chemical surface modifications and/or modifications using metal coatings and their impact on bacterial adherence and biofilm formation in vivo, this systematic review has delved into the nuanced realm of nanotechnology, innovative coatings, and surface treatments. However, critical research gaps always persist, as there is a limited number of in vivo studies with a high risk of bias. Moreover, the limitations encountered in the in vivo animal studies, such as relatively limited sample sizes, heterogeneity in oral microbiomes, variations in host immune responses, along with discrepancies in experimental approaches and bacterial analysis methods, emphasize the need for cautious interpretation of the results of the studies included in this project.

In conclusion, the quest for the ideal implant surface involves a delicate balance between ensuring biocompatibility, fostering osseointegration, and mitigating the risk of bacterial colonization. This dynamic plays a pivotal role in preventing peri-implantitis and guaranteeing the long-term success of dental implants. Establishing standardized testing protocols and methodologies would help when comparing results across different studies and developing consensus on the most effective treatments. Extensive clinical studies are necessary to assess the antibacterial efficacy, durability, and safety of antibacterial coatings in real-world conditions. Research should also focus on developing cost-effective manufacturing processes for antibacterial coatings. In that context, interdisciplinary efforts and collaboration can lead to innovative solutions and accelerate the translation of research into practice.

## Figures and Tables

**Figure 1 antibiotics-13-00908-f001:**
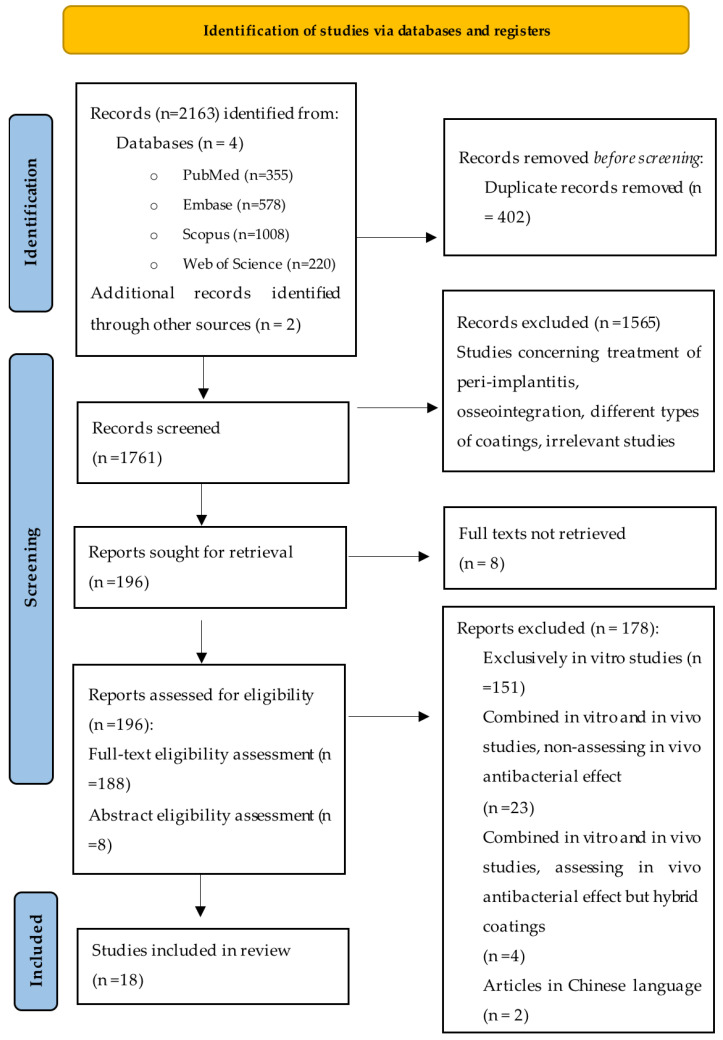
PRISMA flowchart illustrating the study selection process. The diagram details the number of records identified, screened, excluded, and the reasons for exclusion, as well as the final number of studies included in the systematic review [84].

**Figure 2 antibiotics-13-00908-f002:**
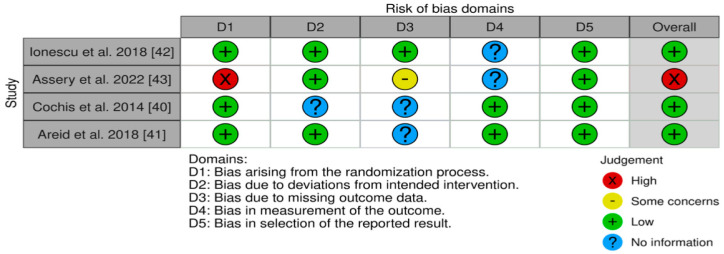
Risk of bias for the in vivo in human studies [86,87].

**Table 1 antibiotics-13-00908-t001:** Study characteristics of the included studies.

Studies	In Vivo Model	Location of Implant Placement	Surface Modification	Procedure of Infection Set-Up	Follow-Up	Antibacterial Efficacy Tests	Conclusions
	Animals						
	Rats						
Pan et al., 2024 [38]	30 eight-week-old male SD rats (Sprague–Dawley)	In the alveolar bone in front of the maxillary first molar.	Anodization (5 h, 50 V), annealing treatment (450 °C, 2 h) and hydrothermal treatment to construct SrTiO_3_/TiO_2_ nanotubes (STNT) and and Al-STNT. Ultrasound treatment every 2 days with a power of 1.5 W cm^−2^ for 5 min, 7 treatments over 2 weeks.	After one month of implantation, the peri-implantitis model was constructed by a silk 4-0 ligature around the neck of the implant and bacteria injection (5 × 10^7^ CFU/mL P. gingivalis and 5 × 10^7^ CFU/mL F. nucleatum) every other day around the thread, for 2 weeks.	2 weeks	Hematoxylin and eosin (H&E) staining Giemsa staining method Bacterial culture.	“After ultrasound treatment, bacteria stained in dark purple were abundantly distributed in the epithelial and submucosal layers in the Ti group. The bacteria in the mucosa of the other experimental groups gradually decreased, and almost no bacteria stained by Giemsa were observed in the last Al-STNT group.” “In vivo experiments proved that the ultrasound-treated Al-STNT implant could effectively resist bacteria-induced peri-implantitis.”
Wen et al., 2023 [34]	Eight male SD rats	The buccal, mesial root alveolar fossa of the first maxillary molar after tooth extraction.	ZnO nanoparticle loaded mesoporous TiO_2_ coatings (nZnO/MTC-Ti) via the evaporation-induced self-assembly method (EISA) and one-step spin coating.	Porphyromonas gingivalis (Pg) at a density of 10^4^/mL.	14 days	Two groups of bacterial culture under aerobic and anaerobic conditions, respectively, for 24 h. Finally, the culture solution was photographed, and its optical density value was read at 600 nm.	“The synergistic effects of MTC and ZnO NPs on nZnO/MTC-Ti could control the Zn^2+^ release at long-term, steady rates while avoiding overdose cytotoxicity without triggering excessive ROS or inducing cell apoptosis.” “Compared to Ti, MTC-Ti also exhibited mild bacterial resistance, suggesting the enhanced inhibitory effects of the mesoporous structure on bacterial adhesion.”
Lu et al., 2022 [30]	Nine rats	The dorsal area of each rat.	Cu doped TiO_2_ (TiO_2_-Cu) films were prepared on Ti by magnetron sputtering and subsequent annealing. Irradiation with 808 nm (0.8 W cm^− 2^) NIR light for 20 min.	50 μL of *S. mutans* (1 × 10^5^ CFU/mL) was dropped in for 5 min.	2 days	Spread plate technique Photothermal images of Ti, TiO_2_-0Cu and TiO_2_-1Cu after 10 min and 20 min in vivo irradiation of NIR light.	“Upon the irradiation of 808 nm NIR light, TiO_2_-1Cu film with low Cu content also showed favorable in vivo antibacterial activity due to the synergistic effect of photothermal effect and Cu.” “Large number of bacteria still remained on the agar plates of Ti and TiO_2_-0Cu groups, while only a few bacteria were found on the agar plates of TiO_2_-1Cu group, and the antibacterial efficiency reached 96%, showing the excellent in vivo antibacterial performance.”
Wang et al., 2020 [29]	10 male SD rats (Sprague–Dawley)	The dorsal area of each rat.	Ti ZnO nanospheres Ti ZnO nanorods Ti ZnO nanorods−nanospheres hierarchical structure (NRS) Ti-Zr ZnO nanospheres Ti-Zr ZnO nanorods Ti-Zr ZnO nanorods−nanospheres hierarchical structure (NRS)	A 100 μL injection of the bacterial suspension of *S. aureus* was injected to the implant position once a day for 2 weeks.	2 weeks	Bacterial culture/ plate colony counting method	“ZnO nanorods and ZnO nanospheres both had certain levels of antibacterial property, while ZnO NRS had the best antibacterial effect.” “The experimental results exhibited that these doubled-layered structures had great antibacterial activity, good stability, and low toxicity.”
Yang et al., 2019 [28]	45 male SD rats	The femoral medullary cavity in the left knee.	Magnesium (Mg)-incorporated nanotube-modified titanium implants (NT-Mg) by electrochemical anodization and hydrothermal treatment NT-Mg3/3 h.	50 μL of PBS containing methicillin-resistant *Staphylococcus aureus*, 5378 [MRSA], ATCC43300 at a concentration of 1 × 10^6^ CFUs/mL was injected into the medullary cavity using a micropipette.	3, 4, and 35 days	X-ray Micro-CT Histopathological analysis Giemsa staining was used to assess bacterial burden in decalcified histological transverse sections.	“The NT group exhibited slightly reduced radiographic scores compared with the Ti group at day 35 post-surgery (*p* < 0.05), indicating that the nanotubular structure itself also exhibited slight anti-infection potential in vivo.” “Histopathological scores in the Ti and NT groups were all significantly increased compared with the NT-Mg group (*p* < 0.01). The total scores in the NT group were also slightly reduced compared with the Ti group (*p* < 0.05), indicating alleviated bone infection in the NT group.”
Tran et al., 2019 [31]	SD rats, number?	Two femurs for each rat, femoral medullary cavity.	Two-hole 1.5 mm titanium plates and screws, titanium substrates coated at a density of approximately 6 × 10^6^ particles/mm^2^ Se NP through surface-induced nucleation-deposition.	10^5^ CFU for MRSE and 10^2^ CFU for MRSA were injected in 100 μL of sterile 0.9% saline on top of the plate.	4 weeks	Scanning electron microscopy SEM imaging Immunostaining for bacteria with rabbit anti-MRSA and mouse anti-*Staph epidermidis*. Colony counting of biofilm bacteria of extracted screws and of surrounding tissue after swabbing wound pockets.	“Confocal microscopy imaging of the bacteria within the biofilms showed thick and dense layers on the uncoated plates compared to more individual, separated bacteria and bacterial aggregates on the coated plates.”
Jin et al., 2014 [32]	20 male SD rats (four groups of five)	The left femoral medullary cavity.	Plasma immersion ion implantation of Ag and Zn at 30 kV for 90 min.	20 mL of the bacterial suspension with a concentration of 10^4^ CFUs/mL *S. aureus*.	6 weeks	Bacterial culture Radiographic examination (osseous destruction) Histological evaluation (Giemsa staining)	“The bacterial growth on the Ag-PIII and Zn/Ag-PIII groups on agar plates is significantly reduced and the corresponding TSB cultures are negative (clear appearance) disclosing that both Ag-PIII and Zn/Ag-PIII have excellent antibacterial ability in vivo.”
	Mice						
Kuehl et al., 2016 [2]	Female C57BL/6 mice, at least 39? Nineteen mice for *S. epidermis* eight mice for *S. aureus*, at least twelve for combination with DAP/VAN	The dorsal area of each mouse.	Ag-coated titanium-aluminum-niobium (TiAlNb) alloy	5 × 10^2^ to 1 × 10^8^ CFU of bacteria *S. epidermidis* 1457, which were injected directly into the cage percutaneously either immediately after implantation (i.e., perioperative infection) or 2 weeks later (i.e., postoperative infection). *S. aureus* SA113 in a perioperative infection using the minimal infective dose of 1 × 10^3^ CFU per cage.	2, 6, 9, and 14 days for *S. epidermidis*	Samples of TCF (tissue cage fluid) on MHA (Mueller Hinton agar) plates.	“The Ag coating prevents an infection with *S. epidermidis* in an inoculum- and time-dependent manner.” “As single agents, neither preoperative DAP nor VAN nor Ag-coated cages were sufficient to prevent a persistent infection with MRSA. Remarkably, in combination with preoperatively applied DAP, Ag coating prevented the growth of planktonic as well as adherent MRSA cells, resulting in a 100% prevention rate.”
	Dogs						
Sun et al., 2023 [27]	Three adult male beagle dogs	The third and fourth premolar and the first molar teeth in both mandibular quadrants 3 months after extraction (six implants—two implants of each group—in No. 1 and 2 beagle dogs and three implants—one implant of each group—(Q3) in No. 3).	Nanophase calcium phosphate embedded to TiO_2_ nanotubes after anodic oxidation, annealing process, and electrochemical deposition on selective laser melting (SLM) titanium substrates.	Oral microbioflora without infection set-up.	8 weeks	Next-generation sequencing (NGS) technology, 16S rRNA gene/RNA sequencing.	“The modified nanostructured titanium surfaces affected the genes associated with microbial metabolism, protein synthesis, bacterial locomotion, localization and the integrity of organism cellular membranes.” “The destruction of bacterial cellular membrane on the NTN surface could be one of the vital mechanisms of antibacterial activities of nanostructured titanium surfaces.”
Liu et al., 2022 [36]	Six male beagle dogs	At mandibular premolar extraction site where implants were placed 3 months after teeth extractions.	TiCu alloy with microstructure of α-Ti + Ti2Cu.	Peri-implant infection model by native oral microbiota (ligature and sucrose-rich diet model).	Oral samples (plaque) collected monthly.	Micro CT Histopathological analysis (hematoxylin and eosin) 16S rRNA gene and metagenomic sequence technology	“Histological score of TiCu group was significantly lower than that of Ti group, which indicated the excellent anti-infection ability of TiCu implant.” “Similarity between the saliva microbial compositions of animals with Ti and TiCu implants in the normal implantation model.” “TiCu implant can still maintain the oral microbiota balance in the infection model.”
Yin et al., 2021 [35]	Labrador dogs	At mandibular premolar extraction site after establishment of the experimental canine periodontitis model, 3 months after extraction.	Micro-arc oxidation (MAO) technology to create Zn/Sr experimental implants. The concentrations of the two ions in the electrolyte were 6 times that of 40 μM Zn^2+^ and 6 mM Sr^2+^.	2-0 sutures were tied around the cervical area of the implant to induce peri-implantitis 4 weeks after implant insertion.	8 weeks	H&E staining Gram staining Immunofluorescence staining of CD3	“Large number of bacteria existed in the soft tissue of the control group, while the experimental group had a good antibacterial ability, which could delay the next stage of inflammation.”
	Rabbits						
Zhao et al., 2024 [37]	?male New Zealand white rabbits	Femur defects of rabbits.	Ti-ZnO, and Ti-ZnP2 The ZnO nanorod arrays were synthesized on the pristine Ti by a hydrothermal method.	*S. aureus* suspension (10^5^ CFU mL^−1^) for 30 min.	2 weeks	Hematoxylin and eosin (H&E) staining Giemsa staining Colony counting method	“Numerous bacteria are found in the Ti group, but fewer bacteria are observed from Ti-ZnO.”
Wang et al., 2019 [39]	24 healthy New Zealand big white rabbits (two groups of twelve)	Back muscle of the rabbit.	Ti-Cu sintered alloy with 10wt% Cu (Ti-10Cu).	20 μL of the bacterial suspension with a concentration of 10^5^ CFUs/mL *S. aureus* strain ATCC 6538.	1, 4, 7, and 14 days	Bacterial culture Hematoxylin and eosin (H&E) staining.	“The colonies number in the Ti-10Cu group was significantly less than the number in the cp-Ti group at all investigated intervals although same amount of *S. aureus* was added in the two groups during the surgical procedure (*p* < 0.01).”
	Pigs						
Mathew et al., 2021 [33]	15 adult White Yorkshire Desi female pigs (50% crossbred)	Titanium plates mounted on acrylic were attached to the pigs’ teeth.	Microtexturing and hydrothermal treatment of the commercially available sand blasted acid etched Ti COM implant to create micro-nano textured Ti implant SAN and nanotextured Ti implant TNL.	Oral microbioflora without infection set-up.	2 days	Gram staining, colony counting.	“Microbial attachment was above 5 × 10^5^ CFU cm^−2^ for the commercial implant, while SAN and TNL implants revealed a significantly low microbial count of 1 × 10^5^ CFU cm^−2^ and 0.5 × 10^5^ CFU cm^−20^, respectively.”
	Humans						
Assery et al., 2022 [43]	Four systemically healthy, ≥18-year-old, nonsmoking subjects.	Custom-made acrylic stents on maxillary arch.	Extraoral surface decontamination with Er: YAG 2940 nm.	Oral microbioflora without infection set-up.	12 h overnight (9 p.m. to 9 a.m.)	Multiphoton confocal laser scanning microscopy and Flu-oView software FV1000were used to evaluate and capture the biofilm 3D structure and the live/dead bacteria ratio. Computational analyses of confocal biofilm images.	“Er:YAG laser treatment of titanium implant surfaces does not significantly affect early biofilm formation in the oral cavity.”
Ionescu et al., 2018 [42]	10 subjects (seven women, three men; aged 20–32 years old with a mean age of 22.5 years).	Individual mandibular thermoformed acrylic customized tray. Three half-implants, one for each experimental surface, were fixed horizontally on each buccal side (three on the right and three on the left).	Laser-microtextured in 136 s, Nd: YAG source diode pumped solid state (DPSS) laser (355 nm wavelength).	Oral microbioflora without infection set-up.	2 days	MTT assay, confocal laser scanning microscopy (CLSM), scanning electron microscopy (SEM) and energy-dispersive X-ray spectroscopy (EDS)	“Machined and laser-treated surfaces were less colonized than grit-blasted ones, while no significant differences were identified between machined and laser-treated surfaces” “Laser-created microtopography can reduce biofilm formation, with a maximum effect when the surface is blasted orthogonally by the laser beam.”
Areid et al., 2018 [41]	10 healthy, nonsmoking adult volunteers (six males, four females, mean age 39.7 years, from 25 to 56 years)	Titanium discs were attached on subjects’ buccal surfaces of their maxillary molars with flowable composite resin.	Nanoporous titanium dioxide surfaces (TiO_2_) obtained by the hydrothermal HT coating method. Half of the discs: UV light for 60 min under ambient conditions using a 36 W puritec HNS germicidal ultraviolet lamp (Osram GmbH; Germany), with a dominant wavelength of 254 nm.	Oral microbioflora without infection set-up.	24 h	Colony counting of plaque and bacterial samples.	“The plaque samples of noncoated groups (NC and UVNC) showed more often *S. mutans* in the biofilms than the coated hydrothermal groups (HT and UVHT) with the number of colonized surfaces equal to seven and three, respectively.” “Hydrothermally induced nanoporous TiO_2_ surfaces inhibited *S. mutans* adhesion and decreased biofilm formation when compared with noncoated titanium alloy.”
Cochis et al., 2015 [40]	Seven subjects (four males and three females; aged 20– 27 years, mean age 24 years)	Oral appliances containing six specimens (1 mm diameter)	Addition of silver (Ag) and gallium (Ga) by electrochemical surface modification using the anodic spark deposition (ASD) method. AgNPs, AgNO_3_, Ga(NO_3_)_3_ appropriately mixed with L-cysteine and oxalic acid dehydrate as chelating agents.	Oral microbioflora without infection set-up.	24 h	Colony counting method Colorimetric assay 2,3-bis (2-methoxy-4-nitro-5-sulfophenyl)-5-((phenyl amino) carbonyl)-2H-tetrazolium hydroxide (XTT; Sigma-Aldrich)	“Gallium-based samples showed the best bactericidal activity among the antibacterial treatments developed. In fact, the values of CFU/mm^2^ were reduced by about 48% with GaCis (Ga(NO_3_)_3_ with chelating agent l-Cysteine) and by 40% with GaOss (Ga(NO_3_)_3_ with chelating agent oxalic acid), when compared with the control SiB-Na (alkali treatment), while the samples containing silver displayed a bacterial inhibition of about 30% (AgCis—AgNO_3_ with chelating agent l-Cysteine) to 34% (AgNPs—Ag nanoparticles).” “Bacteria viability on gallium-treated samples as measured by the inhibition ratio was in a range between 27% (GaOss) and 35% (GaCis) compared to controls. Silver samples confirmed CFU results, but with a slightly lower inhibition ratio than gallium.”

**Table 2 antibiotics-13-00908-t002:** Implant characteristics of the included studies.

Study	Implant Number	Implant Dimensions	Implant Shape	Surface Structure of Tested Implants	Comparison	In Vivo Antibacterial Result
				Ag modified		
Kuehl et al., 2016 [2]	At least 39 implants?	8.5 × 1 × 30 mm	Cylindrical cages (so-called “tissue cages”; 8.5 by 1 by 30 mm	Ag-coated titanium-aluminum-niobium (TiAlNb) alloy	Uncoated TiAlNb alloys	“Ag coating prevents an infection with *S. epidermidis* in an inoculum- and time-dependent manner.” “As single agents, neither preoperative DAP nor VAN nor Ag-coated cages were sufficient to prevent a persistent infection with MRSA. Remarkably, in combination with preoperatively applied DAP, Ag coating prevented the growth of planktonic as well as adherent MRSA cells, resulting in a 100% prevention rate.”
				Zn modified		
Zhao et al., 2024 [37]	? implants (Three groups: Ti, Ti-ZnO, and Ti-ZnP2 rods)	Ø 3 mm L: 5 mm	Rods	Ti-ZnO, and Ti-ZnP2 The ZnO nanorod arrays were synthesized on thepristine Ti by a hydrothermal method.	Pure Ti implants	“Numerous bacteria are found in the Ti group, but fewer bacteria are observed from Ti-ZnO.”
Wen et al. 2023 [34]	16 implants	Ø 1.6 mm L: 3.5 mm	Oral implants	ZnO nanoparticle-loaded mesoporous TiO_2_ coating-titanium (nZnO/MTC-Ti)	Pure titanium (Ti), mesoporous TiO_2_ coating-titanium (MTC-Ti), ZnO nanoparticle-loaded-titanium (nZnO-Ti).	“Compared to Ti, MTC-Ti also exhibited mild bacterial resistance, suggesting the enhanced inhibitory effects of the mesoporous structure on bacterial adhesion. Thus, nZnO/MTC-Ti achieved long-term antibiosis success in vivo through the MTC structure regulating Zn^2+^ release. Sustained Zn^2+^ release created lasting antibacterial environment and desirable inhibitory effects.”
Wang et al., 2020 [29]	20 (n in each group?)	8 × 8 × 1 mm	Foils/slices	Ti-ZnO nanorods, Ti-ZnO nanospheres, Ti-ZnO NRS Ti-Zr-ZnO nanorods, Ti-Zr-ZnO nanospheres, Ti-Zr-ZnO NRS	Ti, Ti-Zr	“The bacteria on the surfaces of samples with ZnO nanorods and ZnO nanospheres modification were less than that on naked Ti or Ti−Zr implants, and the samples with ZnO NRS modification had the least number of bacteria.”
Jin et al., 2014 [32]	20 implants	Ø 2 mm L: 7 mm	Cylinder	Zn-PIII, Ag-PIII, and Zn/Ag-PIII	Pure Ti	“Both Ag-PIII and Zn/Ag-PIII have excellent antibacterial ability in vivo.” The small degree of bacterial growth on the roll-over cultures, absence of neutrophils (and presence of spindle-shaped fibroblast cells indicate the excellent antibacterial ability of the Ag-PIII group in vivo.”
				Cu modified		
Lu et al., 2022 [30]	Nine implants (n = 3, 3 groups)	Ø 5 mm L: 10 mm	Oblong	TiO_2_-1Cu	Ti, TiO_2_-0Cu	“A large number of bacteria still remained on the agar plates of Ti and TiO_2_-0Cu groups, while only a few bacteria were found on the agar plates of TiO_2_-1Cu group, and the antibacterial efficiency reached 96%, showing the excellent in vivo antibacterial performance.”
Liu et al., 2022 [36]	24 implants (12 Ti and 12 TiCu)	Ø 3.6 mm L: 8 mm	Oral implants	TiCu alloy with microstructure of α-Ti + Ti2Cu, sandblasted with large grift and acid etching (SLA) treatment.	Pure Ti implants	“Histological score of TiCu group was significantly lower than that of Ti group, which indicated the excellent anti-infection ability of TiCu implant” “Similarity between the saliva microbial compositions of animals with Ti and TiCu implants in the normal implantation model.” “TiCu implant can still maintain the oral microbiota balance in the infection model.”
Wang et al., 2019 [39]	24 implants (Two groups of twelve)	10 mm in length, 3 mm in width and 1 mm in thickness.	Oral implants	Ti-10Cu	Commercial pure cp-Ti	“Ti-10Cu alloy could kill the bacteria shortly after 1 day implantation and nearly kill all bacteria after 4 days while no such function could be found in the case of the cp-Ti.” “Many colonies were found in the cp-Ti group at all intervals, but only several bacteria colonies could be found in the Ti-Cu group after 1–4 days postimplantation and no bacteria colonies after 7 days postimplantation.”
				Mg modified		
Yang et al., 2019 [28]	45 implants (n = 15, 3 groups)	Ø 2 mm L: 15 mm	Rods	Titanium substrates coated with Mg incorporated TNTs (Electrochemical anodization and hydrothermal treatment to create NT-Mg3).	(1) Pure titanium implant group (Ti) n = 15; (2) TNT-coated titanium implant group (NT) n = 15.	“The radiographic scores recorded in the NT-Mg group were significantly reduced compared with that in the Ti and NT groups from days 14 to 35 after implantation (*p* < 0.01). The NT group exhibited slightly reduced radiographic scores compared with the Ti group at day 35 post-surgery (*p* < 0.05).” “The histopathological scores in the Ti and NT groups were all significantly increased compared with the NT-Mg group (*p* < 0.01). The total scores in the NT group were also slightly reduced compared with the Ti group (*p* < 0.05).”
				Se modified		
Tran et al., 2019 [31]	?	Width 1.5 mm plate	Plates and screws	Selenium NP coated titanium substrates	Non-coated titanium substrates	“Se NP coatings strongly inhibited biofilm formation on the implants and reduced the number of viable bacteria in the surrounding tissue.”
				Ga modified		
Cochis et al., 2015 [40]	42 titanium discs (six groups: Ti, SiB-Na, AgCis, GaCis, GaOss, AgNPs)	Ø 12 mm 0.5 mm thickness	Discs	AgCis, GaCis, GaOss, AgNPs	Ti control, SiB-Na (alkali treatment).	“CFU counts showed a reduction in bacterial colonies on the treated samples compared with the controls. Silver-coated samples resulted in 30–34% decrease in bacterial colonies compared to the controls.” A reduction in the bacterial metabolic activity was observed in gallium- and silver-treated specimens, with the best inhibition ratio in the gallium specimens (27–35%), as confirmed by XTT analysis.”
				Combined metals		
Pan et al., 2024 [38]	30 implants	Ø 1.8 mm L: 3.5 mm	Oral implants	TNT STNT Al-STNT	Ti	“Introducing Al^3+^ ions doping in SrTiO_3_ created crystal distortion, which leads to enhanced sonocatalytic efficiency by oxygen vacancies and piezoelectric properties. This increases the generation of ROS induced by ultrasound, making Al-STNT a promising candidate for antibacterial applications.”
Yin et al., 2021 [35]	12 implants	Ø 2.0 mm L: 10 mm	Rods	Micro-arc oxidation (MAO) technology to create Zn/Sr experimental implants MAO-6.	MAO	“A large number of bacteria existed in the soft tissue of the control group, while the experimental group had a good antibacterial ability, which could delay the next stage of inflammation.”
				Physical/Chemical modifications		
Sun et al., 2023 [27]	15 (Five titanium implants of each group). 12 implants (MP = 3, NT = 5, NTN = 4) were recruited in this study.	Ø 4.5 mm L: 13 mm	-	Nanophase calcium phosphate embedded to TiO_2_ nanotubes (NTN).	(i) Mechanical polishing (MP), (ii) TiO_2_ nanotubes (NT).	“The nanostructured titanium had little effect on the community composition of the sub-mucosal and supra-mucosal microbiota established on implant surfaces. No significant difference of diversity and species richness were found among all groups.”
Assery et al. 2022 [43],	24 titanium discs (two groups of twelve) Three titanium discs were discarded during processing.	Ø 5 mm 1 mm in thickness	Discs	Titanium discs that underwent extraoral surface decontamination with Er: YAG 2940 nm.	Untreated titanium discs (12)	“Er: YAG laser treatment of titanium implant surfaces does not significantly affect the early biofilm formation in the oral cavity.”
Mathew et al., 2021 [33]	45 implants (n = 15, 3 groups)	Ø 4.2 mm L: 12 mm	Cylinder	Micro-nano textured Ti (SAN) and nanotextured Ti (TNL).	Commercially available sand blasted and acid etched Ti (Nano polished) (COM).	“SAN and TNL surfaces with their superhydrophilic character and high surface energy could reduce bacterial attachment by nearly 90% in vivo, as compared to the microscale surface of COM.” “The micro-nano and nanotextured Ti dental implants in our study demonstrate anti-adhesive properties, resulting in reduced bacterial activity, which in turn can inhibit bacterial film formation.”
Ionescu et al., 2018 [42]	30 half-implants	Implant model “Milano” Ø 4.0 mm, L: 9.0 mm	Oral implants	Laser-treated laser-microtextured in 136 s	Machined, grit-blasted	“Machined and laser-treated surfaces were less colonized than grit-blasted ones, while no significant differences were identified between machined and laser-treated surfaces. When the beam blasts the titanium surface at a different angle, as on the inclined portion of the threads, this effect is lost. “
Areid at al., 2018 [41]	40 Titanium (Ti-6Al-4V) alloy discs (4 groups)	Ø 4 mm 1 mm thickness	Discs	UV-treated noncoated titanium alloy (UVNC), hydrothermally induced TiO_2_ coating (HT), UV-treated titanium alloy with hydrothermally induced TiO_2_ coating (UVHT).	Noncoated titanium alloy (NC)	“Noncoated Ti-6Al-4V (NC) surfaces showed over 2 times more *S. mutans* in the early biofilm when compared with the hydrothermally (HT) induced nanoporous TiO_2_ surface. The numbers of colonized surfaces on NC and HT surfaces were equal to 7 and 3, respectively.”

**Table 3 antibiotics-13-00908-t003:** Risk of bias assessment results based on SYRCLE’s risk of bias tool for the included animal studies [88].

*#*	Signalling Question	Wang et al., 2020 [29]	Sun et al., 2023 [27]	Lu et al., 2022 [30]	Mathew et al., 2021 [33]	Yang et al., 2019 [28]	Jin et al., 2014 [32]	Wen et al., 2023 [34]	Kuehl et al., 2016 [2]	Tran et al., 2019 [31]	Yin et al., 2021 [35]	Liu et al., 2021 [36]	Zhao et al., 2024 [37]	Wang et al., 2019 [39]	Pan et al., 2024 [38]
1	Was the allocation sequence adequately generated and applied? (Selection)	Unclear	Unclear	Unclear	Unclear	Unclear	Unclear	Unclear	Unclear	Unclear	Unclear	Unclear	Unclear	Unclear	Unclear
2	Were the groups similar at baseline or were they adjusted for confounders in the analysis? (Selection)	Low	Low	Unclear	Low	Low	Low	Low	Low	Low	Unclear	Low	Low	Low	Low
3	Was the allocation adequately concealed? (Selection)	High	High	Unclear	Low	Unclear	Unclear	Unclear	Unclear	Unclear	Unclear	Low	High	Unclear	Unclear
4	Were the animals randomly housed during the experiment? (Performance)	Unclear	Unclear	Unclear	Unclear	Unclear	Low	Unclear	Unclear	Low	Unclear	Unclear	Unclear	Unclear	Unclear
5	Were the caregivers and/or investigators blinded from knowledge which intervention each animal received during the experiment? (Performance)	High	High	High	Unclear	Low	Unclear	Unclear	High	Unclear	Unclear	Low	Unclear	Unclear	Unclear
6	Were animals selected at random for outcome assessment? (Detection)	High	High	High	Unclear	Unclear	Low	Unclear	Unclear	Unclear	Unclear	High	Unclear	Unclear	Unclear
7	Was the outcome assessor blinded? (Detection)	Unclear	High	High	Unclear	High	Low	Unclear	High	High	Unclear	Low	Unclear	Unclear	Unclear
8	Were incomplete outcome data adequately addressed? (Attrition)	Unclear	Unclear	Unclear	Unclear	High	Unclear	High	Unclear	Unclear	Unclear	Unclear	Unclear	Unclear	Unclear
9	Are reports of the study free of selective outcome reporting? (Reporting)	High	High	High	High	High	High	High	High	High	High	High	High	High	High
10	Was the study apparently free of other problems that could result in high risk of bias? (Other)	High	High	High	High	High	High	High	High	High	High	High	High	High	High

**Table 4 antibiotics-13-00908-t004:** The ARRIVE guidelines 2.0 checklist for the quality assessment of the selected animal studies [89,90].

Items	Wang et al., 2020 [29]	Sun et al., 2023 [27]	Lu et al., 2022 [30]	Mathew et al., 2021 [33]	Yang et al., 2019 [28]	Jin et al., 2014 [32]	Wen et al., 2023 [34]	Kuehl et al., 2016 [2]	Tran et al., 2019 [31]	Yin et al., 2021 [35]	Liu et al., 2022 [36]	Zhao et al., 2024 [37]	Wang et al., 2019 [39]	Pan et al., 2024 [38]
*1. Title*	1	1	1	1	1	1	1	1	1	1	1	1	1	1
*2. Abstract*	1	1	1	1	1	1	1	1	1	1	1	1	1	1
** *INTRODUCTION* **														
*3. Background*	1	1	1	1	1	1	1	1	1	1	1	1	1	1
*4. Objectives*	1	1	1	1	1	1	1	1	1	1	1	1	1	1
** *METHODS* **														
*5. Ethical statement*	1	1	1	1	1	1	1	1	1	1	1	1	1	1
*6. Study design*	1	1	1	1	1	1	1	1	1	1	1	1	1	1
*7. Experimental procedures*	1	1	1	1	1	1	1	1	1	1	1	1	1	1
*8. Experimental animals*	1	1	1	1	1	1	1	1	1	1	1	1	1	1
*9. Housing and husbandry*	0	0	0	1	0	1	0	1	1	0	0	0	0	1
*10. Sample size*	1	1	1	1	1	1	1	1	0	0	1	0	1	1
*11. Allocating animals to experimental groups*	0	0	0	1	1	0	0	0	0	0	0	0	0	0
*12. Experimental outcomes*	1	1	1	1	1	1	1	1	1	1	1	1	1	1
*13. Statistical methods*	1	1	1	1	1	1	1	1	1	1	1	1	1	1
** *RESULTS* **														
*14. Baseline data*	1	1	1	1	1	1	1	1	1	1	1	1	1	1
*15. Numbers analysed*	1	1	1	1	1	1	1	1	0	0	1	1	1	1
*16. Outcomes and estimation*	1	1	1	1	1	1	1	1	1	1	1	1	1	1
*17. Adverse events*	0	0	0	0	0	0	0	0	0	0	0	0	0	0
** *DISCUSSION* **														
*18. Interpretation/scientific implication*	1	1	1	1	1	1	1	1	1	1	1	1	1	1
*19. Generalisability/translation*	0	0	0	1	0	0	0	0	0	0	0	0	0	0
*20.Funding*	1	1	1	1	1	1	1	1	1	1	0	0	0	1
** *TOTAL* **	16	16	16	19	17	17	16	17	15	14	15	14	15	17

## Data Availability

All data are provided with the manuscript.

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
