# Peer review of "Antimicrobial Effects of Metal Coatings or Physical, Chemical Modifications of Titanium Dental Implant Surfaces for Prevention of Peri-Implantitis: A Systematic Review of In Vivo Studies"

_antibiotics, 2024, doi:10.3390/antibiotics13090908_

Round 1

Reviewer 1 Report (Previous Reviewer 1)

Comments and Suggestions for Authors

The submitted manuscript is a systematic review summarizing selected papers on in vivo studies of titanium dental implants in which the effectiveness of their surface modification on antimicrobial effect is described. The systematic review includes 12 new (last 10 years) reports closely related to this topic. Such paper surely will be useful for the professionals involved in the research and development of titanium implants.

I am satisfied with the authors' answers and the improvement of the paper.

I recommend to publish the paper in the Antibiotics journal.

Author Response

Dear Reviewer,

We would like to extend our sincere gratitude for your thoughtful and positive feedback on our manuscript. We are pleased to hear that you found our revisions satisfactory and that you see value in the work we have done. Your recommendation to publish in the Antibiotics journal means a great deal to us, and we appreciate your support throughout this process.

Thank you once again for your time and effort in reviewing our paper.

Best regards,

Maria Gkioka

Reviewer 2 Report (Previous Reviewer 4)

Comments and Suggestions for Authors

Dear authors, thank you for the effort revising the manuscript.
However, you did not address most of the major concerns I had.

Detailed feedback was given in the first round of comments. The major
changes requested have not been addressed, so these are the improvements
they need to apply to their manuscript for enhancing its quality
(PROSPERO registration, complete search equation, better organization of
the Material and Methods and Results sections). All details were
provided in my first review.

Author Response

Dear Reviewer,

Thank you for your continued feedback. However, we would like to respectfully clarify that all of your comments from the first round of revision were thoroughly addressed in our revised submission.

Specifically:

Title: We shortened the title and included "Systematic Review" as per your suggestion.

Results Section: The placement of Tables 1 and 2 has been corrected to appear immediately after their mention in the text.

PRISMA Checklist: This has been completed and presented as supplementary material.

PROSPERO Registration: We made attempts to obtain a PROSPERO registration number before our resubmission in July. However, we were informed by the PROSPERO team that our manuscript was too far along in the process for registration at that stage.

Web of Science: We included Web of Science in our database search as recommended.

Search Equation: We revised the search equation, adding additional terms related to antimicrobial effects and human/animal studies. This was done in accordance with your initial suggestions, and we believe the adjustments made were substantial.

Sections 4.4, 4.5, and 4.6: These sections have been carefully reviewed and revised. We firmly believe that their current placement in the Material and Methods section is appropriate, as they pertain to methodology rather than results.

Reference Style: All references have been checked and corrected to meet the required style guidelines.

Best regards,

Maria Gkioka

Reviewer 3 Report (New Reviewer)

Comments and Suggestions for Authors

This systematic review addresses an important issue in implantology but has some points to be revised.

In Introduction, please replace the expression “the purpose of this project is to investigate…” to “the purpose of this study was to investigate…”

In Results, you stated that 10 studies employing experimental animal models and 4 studies involving human experiments were included. Wouldn't there be 14 studies with animal models?

In Material and Methods, section “4.1 Protocol Development” you stated that “This review was conducted in accordance with the PRISMA”. I suggest replacing the verb “conducted” by “reported”, since PRISMA is a “reporting guideline”.

Why did you limit the search to studies published in the last ten years? Please include a justification for this.

In exclusion criteria, you excluded “any other irrelevant articles”. What did you consider irrelevant articles? How was this criterion applied?

Please report how many reviewers selected each retrieved study and whether they worked independently. Also report how many reviewers collected data from each included study, whether they worked independently, and how you confirmed the data collected by the investigators.

You stated that 8 full texts could not be retrieved, and an eligibility assessment for these was carried out based on the abstract. In the flow diagram of selection process, you described “Title eligibility assessment (n =8)”. Were these 8 studies included for data extraction without reading to the full text? If yes, how did you assess the quality of these studies, and how sure you are that data was completely presented on abstract? In my opinion, if the study’s full text was not retrieved, the study should not be included.

Author Response

Dear Reviewer,

Thank you for your thorough review and for providing such valuable feedback on our manuscript. We greatly appreciate your efforts to help us enhance the quality and clarity of our work.

We are pleased to inform you that we have addressed all of the points you raised:

In Introduction, please replace the expression “the purpose of this project is to investigate…” to “the purpose of this study was to investigate…”

We have revised the expression "the purpose of this project is to investigate…” to “the purpose of this study was to investigate…,” as you suggested.

In Results, you stated that 10 studies employing experimental animal models and 4 studies involving human experiments were included. Wouldn't there be 14 studies with animal models?

The discrepancy in the number of studies has been corrected. The Results section now accurately reflects that 14 studies with animal models were included.

In Material and Methods, section “4.1 Protocol Development” you stated that “This review was conducted in accordance with the PRISMA”. I suggest replacing the verb “conducted” by “reported”, since PRISMA is a “reporting guideline”.

We have replaced the verb "conducted" with "reported" to align with the correct usage of PRISMA as a reporting guideline.

Why did you limit the search to studies published in the last ten years? Please include a justification for this.

We have added a justification for limiting our search to studies published in the last ten years. This was done to ensure that our review reflects the most current research and advancements in the field.

In exclusion criteria, you excluded “any other irrelevant articles”. What did you consider irrelevant articles? How was this criterion applied?

We have addressed this by providing a more detailed explanation in the revised manuscript. (Highlighted in red) Specifically, the term "irrelevant articles" referred to studies within the broader field of dentistry or implantology that did not align with our specific research focus on the antibacterial properties of surface-modified titanium dental implants and their role in peri-implantitis prevention. During the screening process, we excluded studies related to other aspects of implantology, such as biomechanics or osseointegration, as well as articles from other domains of dentistry, including prosthetics, endodontics, and orthodontics. These were deemed outside the scope of our systematic review as they did not directly contribute to answering our research question.

Please report how many reviewers selected each retrieved study and whether they worked independently. Also report how many reviewers collected data from each included study, whether they worked independently, and how you confirmed the data collected by the investigators.

We have clarified this process in our revised manuscript. Specifically, the study selection was conducted by two independent reviewers. Both reviewers individually assessed the titles and abstracts of all retrieved studies to determine their eligibility based on our predefined inclusion and exclusion criteria. In instances where there was disagreement or uncertainty regarding the inclusion of a study, a consensus was reached through discussion between the two reviewers. Regarding data extraction, one investigator was primarily responsible for collecting the relevant data from each included study. To ensure the accuracy and reliability of the extracted data, the second investigator conducted a thorough verification and control of the data.

You stated that 8 full texts could not be retrieved, and an eligibility assessment for these was carried out based on the abstract. In the flow diagram of selection process, you described “Title eligibility assessment (n =8)”. Were these 8 studies included for data extraction without reading to the full text? If yes, how did you assess the quality of these studies, and how sure you are that data was completely presented on abstract? In my opinion, if the study’s full text was not retrieved, the study should not be included.

The flow diagram was corrected accordingly as the eligibility assessment for these 8 articles was carried out based on the abstract. These 8 articles for which the full text could not be retrieved were excluded after an abstract evaluation, as they did not fulfill the inclusion criteria.

Your insightful comments have significantly contributed to the improvement of our manuscript, and we are grateful for your guidance. We believe that the revisions we have made in response to your suggestions have strengthened the quality and clarity of our work.

Thank you once again for your valuable contribution.

Best regards,

Maria Gkioka

Reviewer 4 Report (New Reviewer)

Comments and Suggestions for Authors

This review systematically assessed evidence from “in vivo” studies regarding the antimicrobial efficacy of titanium (Ti) dental implant surfaces following physical/chemical modifications or the application of “various metal element” coatings in preventing bacterial growth associated with peri-implantitis.

Overall this contribution is interesting. Nevertheless, various issues have to be resolved before further consideration.

1.      The authors should revise the title to better reflect the content shown in this manuscript, in which only studies that evaluated the antimicrobial effects of Ti substrate modified with metal elements were enrolled. Studies that used Ti substrates modified with organic or polymer were not enrolled.

2.      Table 1. It is advised that the authors should group these studies according to the animals/humans used in the studies. This would make readers easier to follow through. Similarly, the authors are advised to group the studies shown in Table 2 based on the substrates modified with the same metal element.

3.      Why did the authors not include the studies using the organic coating on the titanium substrates?

Author Response

Dear Reviewer,

Thank you for your thoughtful and constructive feedback on our manuscript. We appreciate your careful review and your suggestions for improving our work.

  1. The authors should revise the title to better reflect the content shown in this manuscript, in which only studies that evaluated the antimicrobial effects of Ti substrate modified with metal elements were enrolled. Studies that used Ti substrates modified with organic or polymer were not enrolled.

We have revised the title of our manuscript to better reflect the content, specifically focusing on studies that evaluated the antimicrobial effects of titanium (Ti) substrates modified with metal elements and other types of physical/chemical modification. We believe this change improves the clarity and accuracy of our manuscript and aligns well with the content presented. Thank you for your valuable input.

  1. Table 1. It is advised that the authors should group these studies according to the animals/humans used in the studies. This would make readers easier to follow through. Similarly, the authors are advised to group the studies shown in Table 2 based on the substrates modified with the same metal element.

As per your suggestion, we have reorganized Table 1 to group the studies according to the type of subjects used (animals or humans), which we believe will indeed help readers to follow the content more easily. Similarly, we have grouped the studies in Table 2 based on the type of surface modification, as advised.

  1. Why did the authors not include the studies using the organic coating on the titanium substrates?

Our decision to exclude studies involving organic coatings was based on the following considerations. Antibacterial metals and alloys provide sustained antibacterial efficacy, as these properties are inherent to the entire alloy. Furthermore, their versatility and ease of production, their potent broad-spectrum antibacterial effects, and the significant advantage of not contributing to the development of drug-resistant strains make them a more suitable option for long-term, consistent antibacterial applications in this study. Our response has been highlighted in red in the revised manuscript.

We hope this explanation clarifies our rationale, and we are grateful for your feedback, which has contributed to strengthening our manuscript.

Kind regards,

Maria Gkioka

Round 2

Reviewer 2 Report (Previous Reviewer 4)

Comments and Suggestions for Authors

Dear authors, as I stated in my two reviews, I find your search strategy poor so the whole review might be biased

This manuscript is a resubmission of an earlier submission. The following is a list of the peer review reports and author responses from that submission.

Round 1

Reviewer 1 Report

Comments and Suggestions for Authors

The submitted manuscript is a systematic review summarizing selected papers on in vivo studies of titanium dental implants in which the effectiveness of their surface modification on antimicrobial effect is described. The systematic review includes 12 new (last 10 years) reports closely related to this topic. Such paper surely will be useful for the professionals involved in the research and development of titanium implants.

Some matters should be improved or complemented:

1) The title has to be changed. A question form is not preferred for scientific papers (it may be accepted for popular-science works) and the title is too long – it has to be shorter which will make it clear. Moreover, a “review” word would be beneficial for the potential readers.

My suggestions are (please try similarly changing the title) :

In vivo studies of titanium dental implants and the effectiveness of their surface modification on antimicrobial effect – a review

OR

In vivo studies of titanium dental implants and the effectiveness of their surface modification on prevention of periimplantitis – a review

2) Line 51 - Surface treatments involve modification (physical, chemical…)

Please add an electrochemical one, especially the mentioned “anodization” is an example of electrochemical modification.

3) The logical structure of some of the sentences is not fully clear; e.g. in the sentence (lines 58-60): “Two categories of coatings are organic (…) and inorganic (…).” Similarly, some phrases are inaccurate, e.g. “no drug-releasing coating” (line 61 or 65) – in this case much better is “drug-free coating” or “drug-free functional coating”. Please, revise the manuscript to remove similar unclear, inaccurate and illogical sentences and phrases. I suggest revising by another person, preferably by an editorial expert.

4) Please, use subscripts and superscripts, e.g. TiO2 (line 119); Zn2+ (line 191); 10^5 (line 203 – and remove the caret ^ here); cm−2 (line 203) and much more. Please, revise the manuscript.

5) Please, use lowercase where they should be used, e.g. “…Poly(ethylene glycol)…” (line 68); “…Zinc (Zn)…” (line 214) “…Selenium” (line 245 / 418). Please, revise the manuscript.

6) Some minor errors:

Line 65 -  “…of drug loaded. [14,23-25].”

Please remove the dot located just before the square bracket.

here (“…poly(ethylene glycol)…”)

Line 97 – “…white Yorkshire…”

Please, use capital letter here (“…White Yorkshire…”)

Line 114 – “…implant surface. (35,36,41)”

Please, put the dot after the bracket / The bracket should be square one (as usual)

Line 215 – “…plasma immersion ion implantation (PII)”

Most probably “(PIII)” should be used here

Lines 238/239 (and 326) – “40μMZn2+ and6mMSr2+”

Please, correct it (unreadable); also in the Table 1

Line 245 – “..with Selenium” ??? Se nanoparticles

It probably should be the end of the sentence (if yes please put a dot).

Line 273 – “…presence of alloying elements, such as silver (Ag+), zinc (Zn2+), and copper (Cu2+)…”

To be more precise it should complemented as follows: “…presence of ions of alloying elements…”

Applying to my critical remarks I recommend to publish it in the MDPI Antibiotics journal after correction/complement the paper in accordance with the remarks.

Author Response

For research article

Response to Reviewer 1 Comments

1. Summary

Thank you very much for taking the time to review our manuscript and for your thorough and valuable feedback. We have carefully addressed each of your recommendations and we have made the necessary corrections to improve the quality of the manuscript. Please find the detailed responses below and the corresponding corrections highlighted in the re-submitted file.

2. Questions for General Evaluation

Reviewer’s Evaluation

Response and Revisions

Does the introduction provide sufficient background and include all relevant references?

Yes

Are all the cited references relevant to the research?

-

Is the research design appropriate?

Yes

Are the methods adequately described?

Can be improved

Are the results clearly presented?

Can be improved

Are the conclusions supported by the results?

Yes

3. Point-by-point response to Comments and Suggestions for Authors

Comment 1: The title has to be changed. A question form is not preferred for scientific papers (it may be accepted for popular-science works) and the title is too long – it has to be shorter which will make it clear. Moreover, a “review” word would be beneficial for the potential readers.

My suggestions are (please try similarly changing the title) :

In vivo studies of titanium dental implants and the effectiveness of their surface modification on antimicrobial effect – a review

OR

In vivo studies of titanium dental implants and the effectiveness of their surface modification on prevention of periimplantitis – a review

Response 1: Thank you for pointing this out. We have shortened the title and we have included the term “systematic review” as well.

Comments 2: Line 51 - Surface treatments involve modification (physical, chemical…)

Please add an electrochemical one, especially the mentioned “anodization” is an example of electrochemical modification

Response 2: We have added the electrochemical one (line 51)

Comment 3:  The logical structure of some of the sentences is not fully clear; e.g. in the sentence (lines 58-60): “Two categories of coatings are organic (…) and inorganic (…).” Similarly, some phrases are inaccurate, e.g. “no drug-releasing coating” (line 61 or 65) – in this case much better is “drug-free coating” or “drug-free functional coating”. Please, revise the manuscript to remove similar unclear, inaccurate and illogical sentences and phrases. I suggest revising by another person, preferably by an editorial expert.

Response 3: We have revised the lines 58-60 and the comment regarding the drug-free coatings and have made the proposed correction. We have revised the whole manuscript in order to ensure the logical structure and clarity of all the statements.

Comment 4: Please, use subscripts and superscripts, e.g. TiO2 (line 119); Zn2+ (line 191); 10^5 (line 203 – and remove the caret ^ here); cm−2 (line 203) and much more. Please, revise the manuscript.

Response 4: We have revised the whole manuscript to incorporate the use of subscripts and superscripts as suggested.

Comment 5: Please, use lowercase where they should be used, e.g. “…Poly(ethylene glycol)…” (line 68); “…Zinc (Zn)…” (line 214) “…Selenium” (line 245 / 418). Please, revise the manuscript.

Response 5: We have revised the manuscript to correct the use of lowercase of as suggested

Comment 6: Some minor errors:

Line 65 -  “…of drug loaded. [14,23-25].”

Please remove the dot located just before the square bracket.

here (“…poly(ethylene glycol)…”)

Line 97 – “…white Yorkshire…”

Please, use capital letter here (“…White Yorkshire…”)

Line 114 – “…implant surface. (35,36,41)”

Please, put the dot after the bracket / The bracket should be square one (as usual)

Line 215 – “…plasma immersion ion implantation (PII)”

Most probably “(PIII)” should be used here

Lines 238/239 (and 326) – “40μMZn2+ and6mMSr2+”

Please, correct it (unreadable); also in the Table 1

Line 245 – “..with Selenium” ??? Se nanoparticles

It probably should be the end of the sentence (if yes please put a dot).

Line 273 – “…presence of alloying elements, such as silver (Ag+), zinc (Zn2+), and copper (Cu2+)…”

To be more precise it should complemented as follows: “…presence of ions of alloying elements…”

Response 6: All corrections have been made as suggested

4. Response to Comments on the Quality of English Language

 There were no comments on the quality of English language.

Reviewer 2 Report

Comments and Suggestions for Authors

This is an interesting review that is worth publishing. The premise is suitable, the paper is well written and the study search strategy is structured and clear. It only needs fewer corrections to be published.

Recommended corrections:

            The abstract, introduction, methods, results, discussion and conclusions are very well written, clear, objective and contains recent refereces.  The review is based only in 12 articles. However, the research strategy was well explained and the results of the articles were carefully described. The study limitations and risk of bias were explored and pointed out. Meta-analysis was not conducted but it was also justified. 

            I would recommend some grammar/abbreviation corrections:

- Line 54: Avoid the abbreviation for “there's” use “there is” instead

- Abbreviations should be done when the word/term is first mentioned in the texts (ECM proteins, TiO2, Zn, Mg, ZnO, Ti, Ti-Zr, TiO2, Se, NT, CFU, TSB….). In the rest of the manuscript the abbreviations can be used without having to write the whole word/term again (e.g lines 196 and 198 with coper). Also remind to use the same abbreviation when referring to the same term (e.g: MCT is equal to MCT-Ti, so standardize it)

- Line 238: Correct to “of Zn and Sr at a specific concentration (6 times that of 40 μM Zn2+ and 6 mM Sr2+

- Line 245: Correct “ Selenium Se” to “Se”.

- Line 278: Correct “hydroxyl radicals OH” to hydroxyl radicals (OH-)

- Be careful with abbreviations. Abbreviations should be done when the word/term is first mentioned in the text (ECM proteins, TiO2, Zn, Mg, ZnO, Ti, Ti-Zr, TiO2, Se, NT, CFU, TSB, nano-ZnO, nano- CuO etc). In the rest of the manuscript the abbreviations can be used without having to write the whole word/term again (e.g lines 196 and 198 with coper). Also remind to use the same abbreviation when referring to the same term (e.g: MCT is equal to MCT-Ti, so standardize it)

- Correct: TiO2 to TiO2, cm-2 to cm-2, 10^5 to 105, Ag+ to Ag+, Zn2+ to Zn2+ and Cu2+ to Cu2+

When referring to an ion is important to mention its electric charge always (e.g Zn2+, Mg2+ etc).

            - Italicize in vivo, in vitro and all bacteria names. 

Author Response

For research article

Response to Reviewer 2 Comments

1. Summary

Thank you very much for taking the time to review our manuscript and for your positive feedback. We are pleased to hear that you found the premise suitable, the paper well-written, and the study search strategy structured and clear and that, in overall, our review is worth publishing. Thank you for all the helpful comments that aim to improve clarity, consistency, and accuracy throughout the manuscript. We have made your recommended corrections to ensure the manuscript meets the highest standards for publication. Please find the detailed responses below and the corresponding corrections highlighted in the re-submitted file.

2. Questions for General Evaluation

Reviewer’s Evaluation

Response and Revisions

Does the introduction provide sufficient background and include all relevant references?

Yes

Are all the cited references relevant to the research?

-

Is the research design appropriate?

Yes

Are the methods adequately described?

Yes

Are the results clearly presented?

Yes

Are the conclusions supported by the results?

Yes

3. Point-by-point response to Comments and Suggestions for Authors

We have proceeded to all your recommended grammar and abbreviation corrections. In particular:

Line 54: We have changed "there's" to "there is".

We have introduced all abbreviations the first time the word/term appears in the text, and we tried to consistently use the abbreviations throughout the manuscript after their first mention and to standardize the use of the same abbreviation when referring to the same term (e.g MCT-Ti instead of MCT)

Line 238: We have corrected the phrase to "of Zn and Sr at a specific concentration (6 times that of 40 μM Zn²⁺ and 6 mM Sr²⁺)".

Line 245: We have corrected "Selenium Se" to "Se".

Line 278: We have corrected "hydroxyl radicals OH" to "hydroxyl radicals (OH⁻)".

We have also revised the whole manuscript in order to correct the use of subscripts and superscripts, like TiO2 to TiO₂, cm-2 to cm⁻², 10^5 to 10⁵, Ag+ to Ag⁺, Zn2+ to Zn²⁺ and Cu2+ to Cu²⁺.

We have ensured that the electric charge is always mentioned when referring to an ion (e.g., Zn²⁺, Mg²⁺).

Finally, we have italicized the terms “in vivo” and “in vitro”, and all bacteria names.

4. Response to Comments on the Quality of English Language

 There were no comments on the quality of English language.

Reviewer 3 Report

Comments and Suggestions for Authors

Due to the diversity of bacteria in the oral environment, peri-implantitis is a common problem in dental implant treatment. Therefore, to prevent peri-implantitis, studies have been focused on endowing implants with antibacterial functions, whether surface treatments or materials themselves, with a large quantity and wide coverage. This article provides a review of the studies on surface treatment endowing implants with antibacterial functions in a representative way. An important question is why there are so many related studies, but almost no clinical applications yet? What should be the main reasons for it? What else needs to be done to drive research towards application? These are all issues that many readers will pay special attention to, and the article should take them into consideration.

Author Response

 For research article

Response to Reviewer 3 Comments

1. Summary

Thank you very much for taking the time to review our manuscript and for valuable feedback.  Your comments have been instrumental in enhancing the quality and relevance of this review. We have carefully addressed your recommendations and we have made the necessary corrections as suggested. Please find the highlighted changes in the re-submitted file in the “conclusion” section.

2. Questions for General Evaluation

Reviewer’s Evaluation

Response and Revisions

Does the introduction provide sufficient background and include all relevant references?

Yes

Are all the cited references relevant to the research?

-

Is the research design appropriate?

Can be improved

Are the methods adequately described?

Yes

Are the results clearly presented?

Can be improved

Are the conclusions supported by the results?

Yes

3. Point-by-point response to Comments and Suggestions for Authors

Comment: Due to the diversity of bacteria in the oral environment, peri-implantitis is a common problem in dental implant treatment. Therefore, to prevent peri-implantitis, studies have been focused on endowing implants with antibacterial functions, whether surface treatments or materials themselves, with a large quantity and wide coverage. This article provides a review of the studies on surface treatment endowing implants with antibacterial functions in a representative way. An important question is why there are so many related studies, but almost no clinical applications yet? What should be the main reasons for it? What else needs to be done to drive research towards application? These are all issues that many readers will pay special attention to, and the article should take them into consideration.

 Response: Thank you for your insightful feedback and for highlighting the important aspects that need to be addressed in our review. We have carefully considered your suggestions and we have revised the manuscript to include the reasons (methodology variability, cost, technical requirements, long-term safety, durability, etc.) behind the abundance of related studies with limited clinical applications for this gap. In addition, we have tried to propose what further steps need to be taken to drive research towards clinical application. These corrections have been applied in the “conclusion” section. Please find the highlighted changes in the re-submitted file.

  1. Response to Comments on the Quality of English Language

There were no comments on the quality of English language.

Reviewer 4 Report

Comments and Suggestions for Authors

Dear authors, you have made a big effort on conducting this systematic review, however, I have found many flaws that make this work unsuitable for publication in this journal. From my point, the search needs to be completely redrawn and it should also be registered prior to commencing the work. I have some suggestions for you to follow if you want to conduct this work properly.

- Title is too long. You should also state "Systematic review" in it.

- Results section: You mention  Tables 1 and 2 here but they do not appear in this part (tables should appear right after the paragraph where they are mentioned.

- Materials and Methods:

- PRISMA checklist should be completed and presented as supplementary material

- You need to register your review in PROSPERO before commencing your work. ID should be stated in your manuscript

- I recommend to use Web Of Science as well as database

- Your search equation is very poor, you need to add more terms, for example, for the antimicrobial block you shold add anti-microbial, bactericidal, antibacterial, anti-bacterial, bacteriostatic, etc. Also, you should add a new block of terms related to human/animal studies.

- Part 4.4 are results, not methodology. You should move this section to the "results" part.

- Part 4.5 are also results, not methodology.

- The same for 4.6 from "Most of the included studies did not give..."

References:

Please revise reference style, some references are not in the correct style.

Author Response

For research article

Response to Reviewer 4 Comments

1. Summary

Thank you for your time, the detailed feedback and for acknowledging the effort put into conducting this systematic review. We much appreciate your constructive suggestions and understand the concerns you have raised. While it is not feasible to completely redraw the search at this point in the process, we assure you that we will carefully consider your comments and recommendations for any future work. Your insights on the need for registering the search prior to commencing the work are particularly valuable, and we will incorporate this practice in upcoming studies to ensure a more rigorous methodology. Please find the detailed responses below.

2. Questions for General Evaluation

Reviewer’s Evaluation

Response and Revisions

Does the introduction provide sufficient background and include all relevant references?

Yes

Are all the cited references relevant to the research?

-

Is the research design appropriate?

Must be improved

Are the methods adequately described?

Must be improved

Are the results clearly presented?

Must be improved

Are the conclusions supported by the results?

Can be improved

3. Point-by-point response to Comments and Suggestions for Authors

Comment 1: Title is too long. You should also state "Systematic review" in it.

Response 1: Thank you for pointing this out. We have shortened the title and have included the term “systematic review” as well.

Comment 2: Results section: You mention Tables 1 and 2 here but they do not appear in this part (tables should appear right after the paragraph where they are mentioned.

Response 2: To avoid any confusion, the sentence “Two overview tables (Tables 1 & 2) have been generated to summarize the fundamental characteristics and key information from the studies included in the analysis.” has been removed from “results” section. Tables 1 & 2 appear at “data extraction and study characteristics” section to ensure logical structure and clarity.

Comment 3: PRISMA checklist should be completed and presented as supplementary material

Response 3: PRISMA checklist is completed and uploaded as supplementary material.

Comment 4: You need to register your review in PROSPERO before commencing your work. ID should be stated in your manuscript

Response 4: As the current review process is already underway and nearing completion, it is not possible to register it in PROSPERO at this stage. We will certainly consider registering future reviews in PROSPERO before commencing the work, and we will ensure the ID is stated in the manuscript as per your recommendation.

Comment 5: I recommend to use Web Of Science as well as database

Response 5: We appreciate your recommendation to use Web of Science along with all the other databases already used. However, after extensive discussion with my supervisor, we concluded that focusing on PubMed, Embase and Scopus databases best suits the needs of this project. Thank you for understanding.

Comment 6: Your search equation is very poor, you need to add more terms, for example, for the antimicrobial block you shold add anti-microbial, bactericidal, antibacterial, anti-bacterial, bacteriostatic, etc. Also, you should add a new block of terms related to human/animal studies.

Response 6: Thank you for your suggestions regarding additional search terms. However, our current search style focused on specific keywords and criteria to narrow down the results. We already found a substantial number of relevant studies, so we aimed to restrict the research to ensure we capture the most pertinent literature within our scope. Including MESH terms could have broaden the search, potentially adding more results that may not have been directly aligned with our current objectives. Thus, we opted to maintain the current focus to manage the volume and relevance of the retrieved articles effectively.

Comment 7: Part 4.4 are results, not methodology. You should move this section to the "results" part.

Response 7: “Identification and study selection”: This section outlines how we identified and selected the studies that formed the basis of our research. It describes the methods and criteria used to search databases, select relevant articles, and exclude irrelevant ones. It details our search strategy and study selection criteria, so that other researchers can assess the validity of our approach. For this reason, we considered this section “methodology” and not “results”.

Comment 8: Part 4.5 are also results, not methodology

Response 8: “Data extraction and study characteristics” is considered part of the methodology because it outlines the systematic approach used to collect and organize data from the selected studies. It demonstrates the procedure followed to extract relevant information, such in vivo model, surface modification, implant characteristics, antibacterial efficacy tests, follow-up periods, outcomes etc. in a standardized and consistent manner. This helps in outcome analysis and supports the research objectives.

Comment 9: The same for 4.6 from "Most of the included studies did not give..."

Response 9: “Risk of bias and assessment of quality for the selected studies” This section describes the methods used to systematically evaluate the potential for bias and the overall quality of the studies included in the review. The paragraph: “The clinical human studies included in the analysis were evaluated...... and only one assessed the generalizability of their findings to evaluate their relevance to the human population.” explains and supports the figure 1 as well as the tables 3 & 4 in “risk of bias and assessment of quality” section.

Comment 10: References: Please revise reference style, some references are not in the correct style.

Response 10: We revised the manuscript to ensure that all references follow the same style throughout the document.

4. Response to Comments on the Quality of English Language

 There were no comments on the quality of the English language.

Round 2

Reviewer 3 Report

Comments and Suggestions for Authors

AS the authors have added appropriate content to the article based on the review comment, the manuscript can be considered for publication in the present state.

Reviewer 4 Report

Comments and Suggestions for Authors

Dear authors, thank you for making some improvements to your manuscript, I am sorry you could not amend the major concerns I raised, which I recommend you to do for the future